# Rickettsial DNA and a *trans*-splicing rRNA group I intron in the unorthodox mitogenome of the fern *Haplopteris ensiformis*

Simon Zumkeller [1], Monika Polsakiewicz[1] & Volker Knoop [1]✉

Plant mitochondrial genomes can be complex owing to highly recombinant structures, lack of gene syntenies, heavy RNA editing and invasion of chloroplast, nuclear or even foreign DNA by horizontal gene transfer (HGT). Leptosporangiate ferns remained the last major plant clade without an assembled mitogenome, likely owing to a demanding combination of the above. We here present both organelle genomes now for *Haplopteris ensiformis*. More than 1,400 events of C-to-U RNA editing and over 500 events of reverse U-to-C edits affect its organelle transcriptomes. The *Haplopteris* mtDNA is gene-rich, lacking only the *ccm* gene suite present in ancestral land plant mitogenomes, but is highly unorthodox, indicating extraordinary recombinogenic activity. Although eleven group II introns known in disrupted *trans*-splicing states in seed plants exist in conventional *cis*-arrangements, a particularly complex structure is found for the mitochondrial *rrnL* gene, which is split into two parts needing reassembly on RNA level by a *trans*-splicing group I intron. Aside from ca. 80 chloroplast DNA inserts that complicated the mitogenome assembly, the *Haplopteris* mtDNA features as an idiosyncrasy 30 variably degenerated protein coding regions from Rickettiales bacteria indicative of heavy bacterial HGT on top of tRNA genes of chlamydial origin.

[1] IZMB – Institut für Zelluläre und Molekulare Botanik, Abteilung Molekulare Evolution, Universität Bonn, Kirschallee 1, 53115 Bonn, Germany. ✉email: volker.knoop@uni-bonn.de

The conserved structure of chloroplast genomes (plastomes) and the stoichiometric dominance of chloroplast (cpDNA) over nuclear and mitochondrial (mtDNA) in total plant nucleic acid preparations have led to a tremendous increase of available complete plastome sequences with the advent of Next Generation Sequencing (NGS) technologies. The number of completed plant mitochondrial genome sequences (mitogenomes) is much lower in contrast, most notably in vascular plants (tracheophytes). This is largely due to the much more variable and complex mitogenome structures in tracheophytes[1–5], which hitherto left complete mitogenome assemblies altogether missing for the large clade of leptosporangiate ferns among the monilophytes.

Complex, recombining mitogenomes clustered with repeat sequences and affected by the lateral invasion of chloroplast DNA, or even by horizontal gene transfer (HGT) from other species, have arisen independently in the four large tracheophyte clades: the angiosperms, the gymnosperms, the monilophytes and the lycophytes. The lycophytes as the evolutionary oldest of the four clades of extant vascular plants reflect this most clearly: The mitogenome in the club moss *Phlegmariurus squarrosus* is a circular DNA with a rich gene complement and even retaining several ancestral gene syntenies of the circular mtDNAs in bryophytes[6]. The mtDNAs of the quillwort *Isoetes engelmannii* or the spike moss *Selaginella moellendorffii*, in contrast, are strongly depauperated in gene content and heavily affected by recombination leading to complex and coexisting arrangements of coding islands embedded between repeated sequences[7,8]. Similarly, moderately compact and gene-rich circular mitogenomes are present in the gymnosperms *Cycas taitungensis*[9] and *Ginkgo biloba*[10]. In contrast, the mitogenomes of *Taxus cuspidata*[11] and *Welwitschia mirabilis*[10] are reduced in gene content and this even despite size increase to nearly 1000 Kbp in the latter case. As reflected from a collection of mitochondrial scaffolds and chromosomal assemblies, respectively, yet much larger mitogenomes are present in the conifer I clade in *Picea* species[12,13] or *Larix sibirica*[14].

Among angiosperms, the mtDNA of the magnoliid *Liriodendron tulipifera* represents an ancestral, gene rich state[15]. Other flowering plant species, however, have complex mtDNAs of enormous sizes exceeding 1000 Kbp, are fragmented into multiple co-existing mitochondrial chromosomes, affected by massive lateral gene transfer (LGT) from the cpDNA or by horizontal gene transfer (HGT) from other species or display combinations of those features to variable degrees. The large mitogenomes exceeding 1 Mbp in the cucumber family[16,17], the diverse multipartite mtDNAs in the genus *Silene*[18] or the mtDNA of the isolated flowering plant *Amborella trichopoda* that is heavily affected by HGT from diverse species[19] are prime examples.

Among monilophytes (ferns *sensu lato* including the horsetails), the second largest group of tracheophytes behind the angiosperms, the situation is less clear since only the two complete mitogenome sequences of *Ophioglossum californicum* (adder's tongue) and the whisk fern *Psilotum nudum* have been determined[20]. Although the two taxa represent sister clades among the eusporangiate ferns, a paraphyletic grade at the base of extant fern taxa, their mitogenomes reflect differences indicating a dynamic evolution of mtDNAs also among the monilophytes. Aside from slight differences in gene and intron complement, the *O. californicum* mtDNA maps as a single circular mtDNA whereas two separate circular chromosomes exist in *P. nudum*[20].

Most of extant fern diversity with more than 10,000 species resides in the leptosporangiate ferns, however. Studies on selected mitochondrial loci among ferns revealed interesting dynamics of group II introns in their mtDNAs[21,22]. The leptosporangiate fern family of Pteridaceae (Polypodiales) proved to be particularly interesting in a study of mitochondrial group II intron gain, loss and coevolution scenarios[22], also with respect to the concomitant evolution of C-to-U and reverse U-to-C RNA editing, which is abundantly present in the endosymbiotic organelles of leptosporangiate ferns[23,24]. The Pteridaceae represent a large fern family comprising some 1150 species in 45 genera, placed into at least five sub-groups and potential sub-families: the cryptogrammoid, the adiantoid-vittarioid, the cheilanthoid-hemionitidoid, the ceratopteroid-parkerioid, and the pteroid ferns. Habitats occupied by these sub-groups are equally diverse, as they range from terrestrial, including epipetric and epiphytic, to even aquatic lifestyles.

We chose *Haplopteris ensiformis* among the epiphytic, vittarioid shoestring ferns for a detailed analysis of its two organelle genomes and transcriptomes. We found a typical, conserved chloroplast genome structure in *H. ensiformis* but identified a highly unorthodox mitogenome characterized by numerous active and inactive repeat sequences and a massive insertion of chloroplast DNA. Transcript maturation of the comparatively rich mitochondrial gene complement, lacking only *ccm* genes for cytochrome maturation, involves splicing of 24 group II and four group I introns and abundant C-to-U and U-to-C RNA editing at nearly 2000 sites. The most surprising novelties of molecular evolution in a plant mitogenome include a unique *trans*-splicing group I intron in the large ribosomal rRNA and, most notably, extended stretches of bacterial, *Rickettsia*-like DNA in the *Haplopteris* mtDNA.

## Results and discussion

Our choice of the shoestring fern *H. ensiformis* as a leptosporangiate fern for complete assembly of both organelle genomes was based on pronounced variability in mitochondrial RNA editing and intron (co-)evolution in the monilophyte family Pteridaceae and the sub-family Vittarioideae in particular[22]. We used next-generation sequencing (NGS) data to assemble the chloroplast (cpDNA) and mitochondrial genomes (mtDNA) of *H. ensiformis*, accompanied by RNA-seq transcriptome analyses to study RNA processing with a special focus on intron splicing and RNA editing. As expected, chloroplast DNA reads dominated in the NGS data with read coverages of ca. 1800–4200 and allowed the straightforward assembly of the *H. ensiformis* cpDNA. The mitochondrial DNA, in contrast, revealed overall lower and much more variable read coverages and its assembly was highly complicated by a multitude of repeats, long intergenic stretches and the insertions of foreign DNA and required multiple independent PCR amplifications for verification and complete assembly.

**The *H. ensiformis* plastome and its well-resolved RNA editome.** The *H. ensiformis* chloroplast DNA (148,805 bp) reveals a typical conserved circular plastome structure with a large (80,986 bp LSC) and a small (20,773 bp SSC) single copy region separated by a pair of inverted repeats (IRs of 23,523 bp each). The chloroplast genome carries 116 genes widely conserved in other land plants, 85 of which encode proteins, including the recently characterized *ycf94* gene, 4 rRNAs and 27 tRNAs (Fig. 1a). Likewise, the *H. ensiformis* cpDNA contains a set of 20 conserved introns. Our accompanying transcriptome analysis confirmed functional splicing for all of them.

One striking structural difference concerns the presence of two mobile ORFs in fern organelles, morffo elements[25], in the chloroplast genome of the related species *Haplopteris elongata* (Fig. 1b). Our data suggest their secondary loss in the now determined cpDNA of *H. ensiformis* rather than an independent gain in *H. elongata* as we will discuss below in the context of the

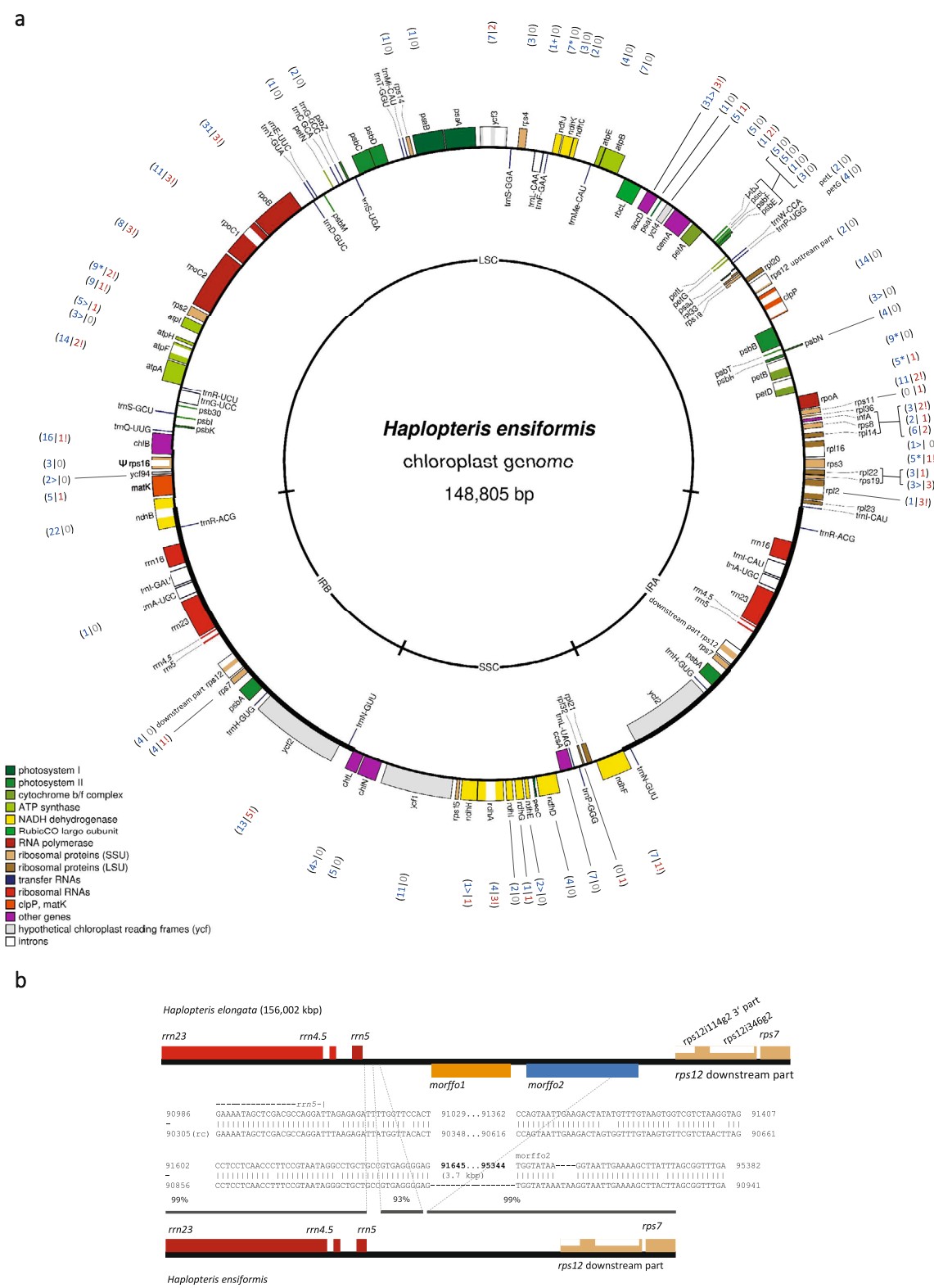

a

*Haplopteris ensiformis*

chloroplast genome

148,805 bp

photosystem I
photosystem II
cytochrome b/f complex
ATP synthase
NADH dehydrogenase
RubicCO large subunit
RNA polymerase
ribosomal proteins (SSU)
ribosomal proteins (LSU)
transfer RNAs
ribosomal RNAs
clpP, matK
other genes
hypothetical chloroplast reading frames (ycf)
introns

b

*Haplopteris elongata* (156,002 kbp)

*Haplopteris ensiformis*

numerous cpDNA inserts that we identified in the now determined *H. ensiformis* mitogenome.

Modern RNA-Seq technologies allow for detection also of low-rate RNA editing sites with reasonable precision and any numbers of reported edits for an organelle transcriptome should nowadays ideally be accompanied by threshold criteria for their detection. Likewise, instead of the frequently used terms

"complete" or "partial" editing, the percentage of detected base conversions by RNA editing should be given for the respective editing sites. The generally high coverage of RNA-Seq reads for the chloroplast transcriptome (mostly above 1000× and here reaching coverages of up to 250,000× for the *psbE* gene in the case of the *H. ensiformis* cpDNA) allows for a detailed evaluation of RNA editing events with high precision, allowing for three digits

**Fig. 1 The Haplopteris ensiformis cpDNA. a** *Haplopteris ensiformis* reveals a typical plant circular plastome structure consisting of a large (LSC) and a small (SSC) single-copy region separated by a pair of inverted repeats (IR) and an expectedly conserved, ancestral gene and intron complement. The genome map was created using OGDRAW[90]. Gene categories are indicated in the legend. Numbers in parentheses indicate the amount of C-to-U (blue) and U-to-C (red) RNA editing for the respective genes. Creations of start or stop codons by C-to-U editing are indicated by symbols ">" and "*" and the removal of stop codons by U-to-C editing is indicated by the exclamation marks, respectively. **b** The cpDNA of *Haplopteris elongata* (accession MH173086) features two "morffo" elements ("mobile ORFs in fern organelles") in the IR region between *rrn5* and the 3'-part of the *trans*-splicing *rps12* gene. Recognizable sequence homologs of morffo2 can presently only be identified in *Cyclosorus interruptus* (accession MN599066, Thelypteridaceae) and *Histiopteris incisa* (accession MH319942, Dennstaedtiaceae) and a homolog of morffo1 (orange) can presently only be found in the distant fern *Hymenophyllum holochilum* (accession MH265124, Hymenophyllales). Only the upstream part of morffo2 (378 bp) is present in the *H. ensiformis* plastome, while a cpDNA insert in its mitogenome contains an extended region of 628 bp.

behind the decimal point. Using stringent criteria for DNA and RNA read qualities (see "Methods"), we could identify 443 sites of chloroplast RNA editing covered by at least 100 RNA reads and RNA editing frequencies of at least 1.0% (Supplementary Data 1).

We here use our previous nomenclature proposal for unequivocal labeling of RNA editing events[26], indicating the affected locus, the nucleotide resulting from C-to-U or reverse U-to-C editing (eU or eC), the position and, for the majority of edits within protein coding regions, the codon meaning before and after the edit (Supplementary Data 1). The careful analysis of RNA editing events contributes crucially to the identification of functional genes in the organelles or the dismissal of others as pseudogenes, notably in plant species like *H. ensiformis* featuring abundant C-to-U and reverse U-to-C RNA editing at the same time. A case in the point is the small reading frame *ycf94/orf51* of hitherto unknown function between *rps16* and *matK*, for which we here find even higher rates of RNA editing at two important sites (ycf94eU2TM and ycf94eU50PL) of 59% and 74%, respectively, than reported previously for other species[27,28]. Vice versa, we consider *rps16* a degenerating pseudogene in *H. ensiformis* as we could not confirm the expected removals of stop codons by reverse U-to-C editing.

The range of observed editing efficiencies extends from 99.1% for the codon sense-changing editing event psbBeU116SL down to 1.0% for edits accDeU657SS (silent), petBi6g2eU345 (intron) or edit rps2eU349R*, which unexpectedly introduces an early stop codon in the *rps2* coding sequence. Such edits at low frequency are likely collateral effects owing to lacking specificity of the chloroplast RNA editing machinery. The same holds true for most codon sense-changing edits that are unexpected (as they do not restore conserved codon identities) and likewise show only inefficient editing with low frequency (Supplementary Data 1).

Most of the detected non-silent RNA editing sites in the chloroplast coding sequences, however, confirm expectations for restoring conserved amino acid positions very well and are efficiently edited and, vice versa, we find low frequencies of editing nearly exclusively in silent or non-coding positions like 5'-or 3'-UTRs (Supplementary Data 1). However, exceptions exist: Prime example for efficient RNA editing events in non-coding regions, which could have been missed altogether in typical RT-PCR-based studies focusing on coding sequences, are cytidine-to-uridine conversions petBi6g2eU478 and rps12i346g2eU80 in the respective introns within *petB* and *rps12* of 94.0% and 90.2%, respectively. Conversely, we observe strikingly low efficiencies in many cases of reverse U-to-C editing removing stop codons, for example rpoC2eC232*Q reconstituting a glutamine codon in the *rpoC2* reading frame is edited with only 17.0% efficiency in the transcript population.

Ten start codons and one stop codon are created by C-to-U editing and altogether 26 stop codons are removed by reverse U-to-C RNA editing in the chloroplast gene transcripts (Fig. 1 and Supplementary Data 1). We here use the *accD* gene as a somewhat less conserved protein coding region as an example for

discussion (Supplementary Fig. 1). Codon-changing edits confirm predictions very well with editing frequencies between 61.0% for accDeC730SP and 94.1% for accDeU779SL. Unpredictable edits in the UTRs or silent position accDeU657SS are edited much less efficiently except for accDeU-1 right upstream of the start codon created by editing (Supplementary Fig. 1b). An intriguing case is editing site accDeU625HY inefficiently edited to 6.4%, for which a histidine or tyrosine is found variably in other taxa. An additional reverse edit is predicted for position accDeC580FL but remained unconfirmed. Note that in most fern cpDNA database entries, just two RNA editing sites are arbitrarily postulated to create an intact *accD* reading frame with a start codon edit in position 2 and to remove a stop codon in position 772.

Given that the organelle transcriptomes of *H. ensiformis*, and especially the mitochondrial transcriptome (see below), proved to be new examples for abundant C-to-U and U-to-C editing, we here use the opportunity to introduce a nomenclature amendment addressing the complex issue of multiple editings in individual codons. We suggest indicating the individual and cumulative effects on codon meaning after and before a pipe symbol (|), respectively, and additional silent edits by an underline symbol (Supplementary Fig. 2). As an example, we consider edits atpBeU1381PL|PS and atpBeU1382PL|PL changing a proline (P) codon identity in the *atpB* transcript. The first position edit takes place with 97.9% efficiency to expectedly reconstitute a conserved serine (S) codon whereas the unexpected additional second position edit causes a change into a leucine (L) codon, although with only 1.7% efficiency. Yet more complex is the example of a CCC proline codon in the *rpoB* gene edited with different frequencies in all three positions. Edit rpoBeU662PF|PL in the second codon position causes the expected change towards a conserved leucine codon with 66.3% efficiency. However, unexpected edit rpoBeU661PF|PS in the first codon position with only 1.2% efficiency would, considered alone, cause a change to serine. In combination with the edit in the second position the codon is changed to a phenylalanine (F) codon (Supplementary Data 1 and Supplementary Fig. 2). Moreover, these two non-silent edits are accompanied by a third position edit with 36% efficiency (rpoBeU663PP_FF), which is silent for any of the four possible codon identities (P, L, S, or F).

**Assembly of the highly complex *H. ensiformis* mitogenome.** The assembly of the *H. ensiformis* mitogenome turned out to be very demanding owing to a combination of several factors, which we will address in separate paragraphs below. The mtDNA sequence reads were not only ca. 10-fold less abundant than those of the cpDNA (on average ca. 150×) but also much more variable in coverage (ca. 70×–600×). We relied on parallel transcriptome analysis to verify authentic native mitochondrial genes characterized by RNA editing. Evident mtDNA contig assemblies ran into numerous repeated sequences, generally represented with higher read coverage or into extended insertions of laterally transferred chloroplast DNA fragments, a typical feature of

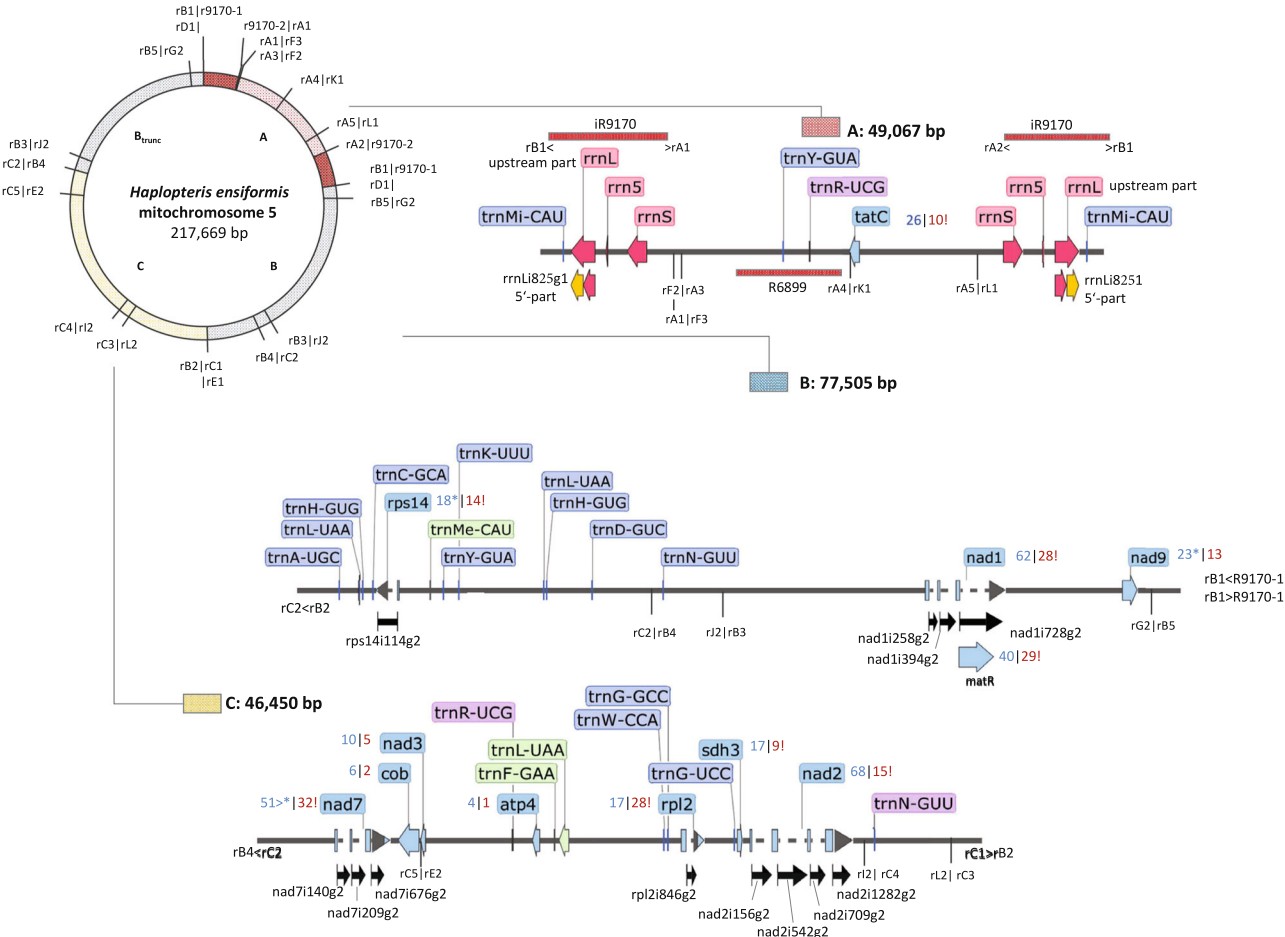

**Fig. 2 The Haplopteris ensiformis mtDNA: contigs A-C and chromosome 5.** *Haplopteris ensiformis* mtDNA contigs A (hatched), B (gray), and C (dotted) can be connected into a circular chromosome of 217,669 bp. Recombination breakpoints are numbered for each contig and preceded by a small "r" with radial lines in the circular map indicating transitions into other contigs, allowing for numerous alternative mtDNA arrangements. Eight further circular chromosomes, as listed on top, connect sequences of chromosome V with nine further mtDNA contigs D-L as shown in Fig. 3. Contig gene maps were created using the SnapGene Viewer software. Native mitochondrial protein coding and tRNA gene sequence are given in lighter and darker blue, respectively, and introns are indicated with black arrows. Numbers next to genes indicate C-to-U (blue) and U-to-C RNA edits (red) with additional symbols indicating removal (!) or creation of stop (*) or start (>) codons, respectively. Introns are indicated with stippled lines and additional black arrows and their standardized labels. Ribosomal RNA genes (here on contig A) are shown in red with the *rrnL* gene featuring a complex gene structure requiring *trans*-splicing via the disrupted group I intron rrnLi825g1 and PSX labels indicate pseudogene fragments shown in gray. The peculiar case of the *trans*-spliced group I intron rrnLi825g1 is highlighted in yellow. Genes for tRNAs of chloroplast or bacterial origin are indicated in green or purple, respectively. For clarity, no other chloroplast or bacterial DNA insertions are shown here. The latter are listed together with annotated features in Supplementary Data 3.

vascular plant mitogenomes. Finally, we found numerous surprising similarities with *Rickettsia*-type bacterial genomes indicating multiple horizontal gene transfer (HGT) beyond the previously identified chlamydial tRNA genes in early branching tracheophytes[29].

Multiple recombination breakpoints allow for a huge spectrum of alternative, and likely co-existing, mitogenome arrangements with variable stoichiometries. For clarity, we chose to assemble twelve mtDNA contigs (A to L, ranging in sizes from 2646 to 77,705 bp) comprising the full mitochondrial sequence complement into nine circular mtDNA chromosomes (chr1 to chr9) as separate GenBank database accessions (OM867545-OM867553). These nine circular chromosomes, however, likely represent only a substoichiometric minority of the truly existing mtDNA molecules owing to the numerous recombination breakpoints (Figs. 2 and 3). In total, 32 recombination breakpoints could be identified. These recombination breakpoints are labeled with "r," the respective contig and consecutive numbers (rA1–rL2). We here display chromosome 5 comprising contigs A–C (Fig. 2) and

chromosomes 1–4 and 6–9, variably integrating contigs D–K into the former (Fig. 3). Among these, contig A features prominently as it is flanked at its ends by repeat R9170 in inverted orientations, the largest repeat sequence that we identified in the *H. ensiformis* mitogenome (Fig. 2). Repeat R9170 carries the upstream part of the ribosomal rRNA gene cluster (Fig. 2), hence distantly reminding of the typical IRs in chloroplast genomes (see Fig. 1).

**Repeats and recombination in the *H. ensiformis* mitogenome.** Other than by the many chloroplast DNA inserts that we will discuss below, the assembly of the *H. ensiformis* mtDNA was complicated by numerous repeated sequences of different sizes. Ancestral mitochondrial gene syntenies are widely eradicated in *H. ensiformis* owing to numerous recombination events—the *rps19-rps3-rpl16* gene continuity on contig F is one remaining exception (Fig. 3). While single-copy sequence contigs with mitochondrial genes had average NGS DNA read coverages

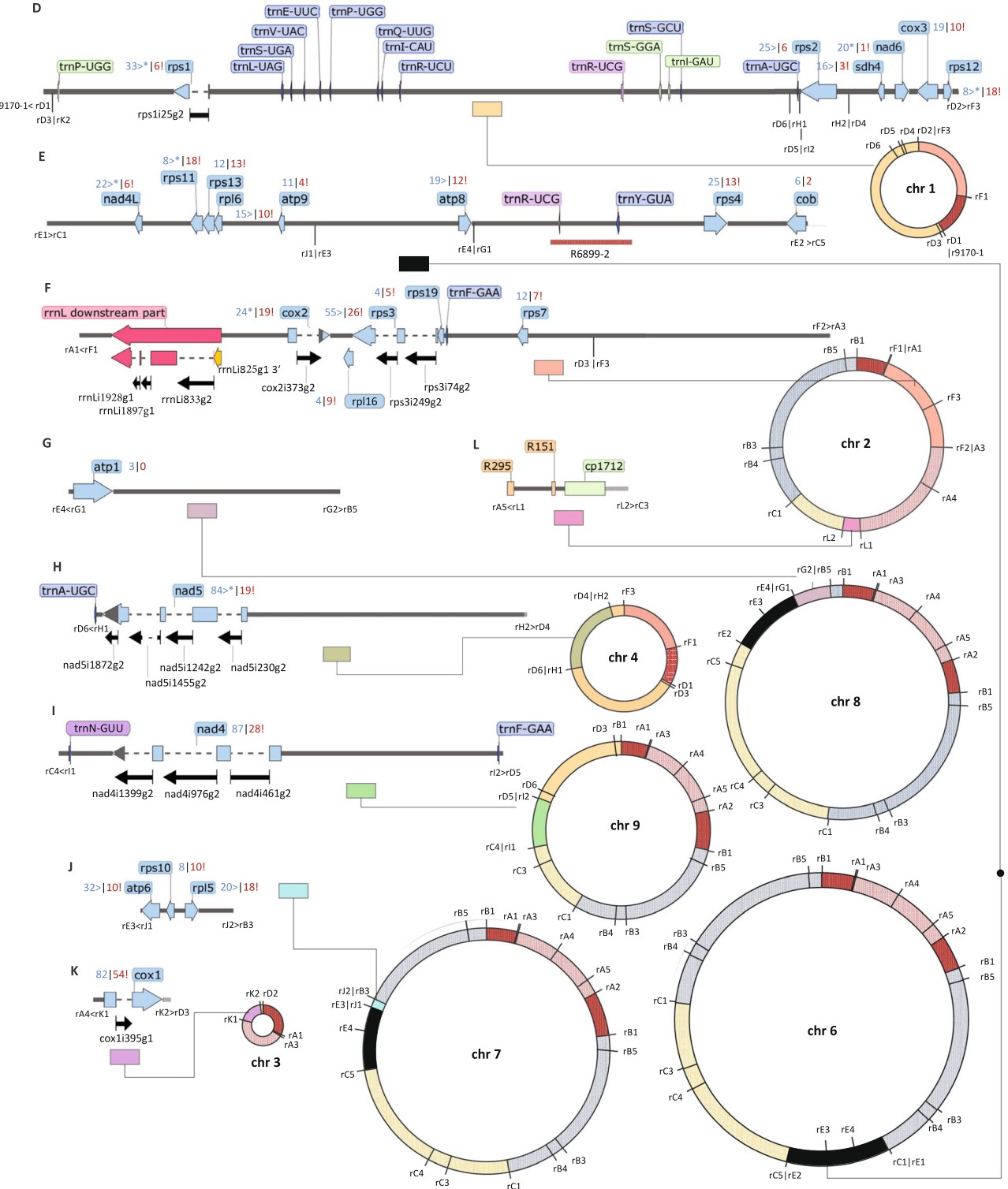

**Fig. 3 The Haplopteris ensiformis mtDNA: contigs D–L and chromosomes 1–4 and 6–9.** The *Haplopteris ensiformis* mtDNA contigs D–L are linked to recombination points in chromosome 5 (Fig. 2) or within themselves, creating further and/or alternative mtDNA arrangements. Recombination endpoints are labeled and the display of contigs and labels for genes and RNA editing events is as in Fig. 2. Possible, circular chromosomal structures chr1 to chr4 and chr6 to chr9 are shown.

around 150-fold (albeit with a broad distribution), identical repeat sequences mostly had coverages exceeding 300-fold. For clarity and discussion, we have labeled and annotated repeats with "R" followed by the number of nucleotides, also in the corresponding GenBank accessions.

We carefully checked on recombinational activity across repeats with a template-switch-avoiding tsa-PCR strategy and examples are shown in Fig. 4. Repeat R596 (Fig. 4a) is a particularly intriguing example as it is not intergenic but shared as an identical sequence present in domains I of group II introns

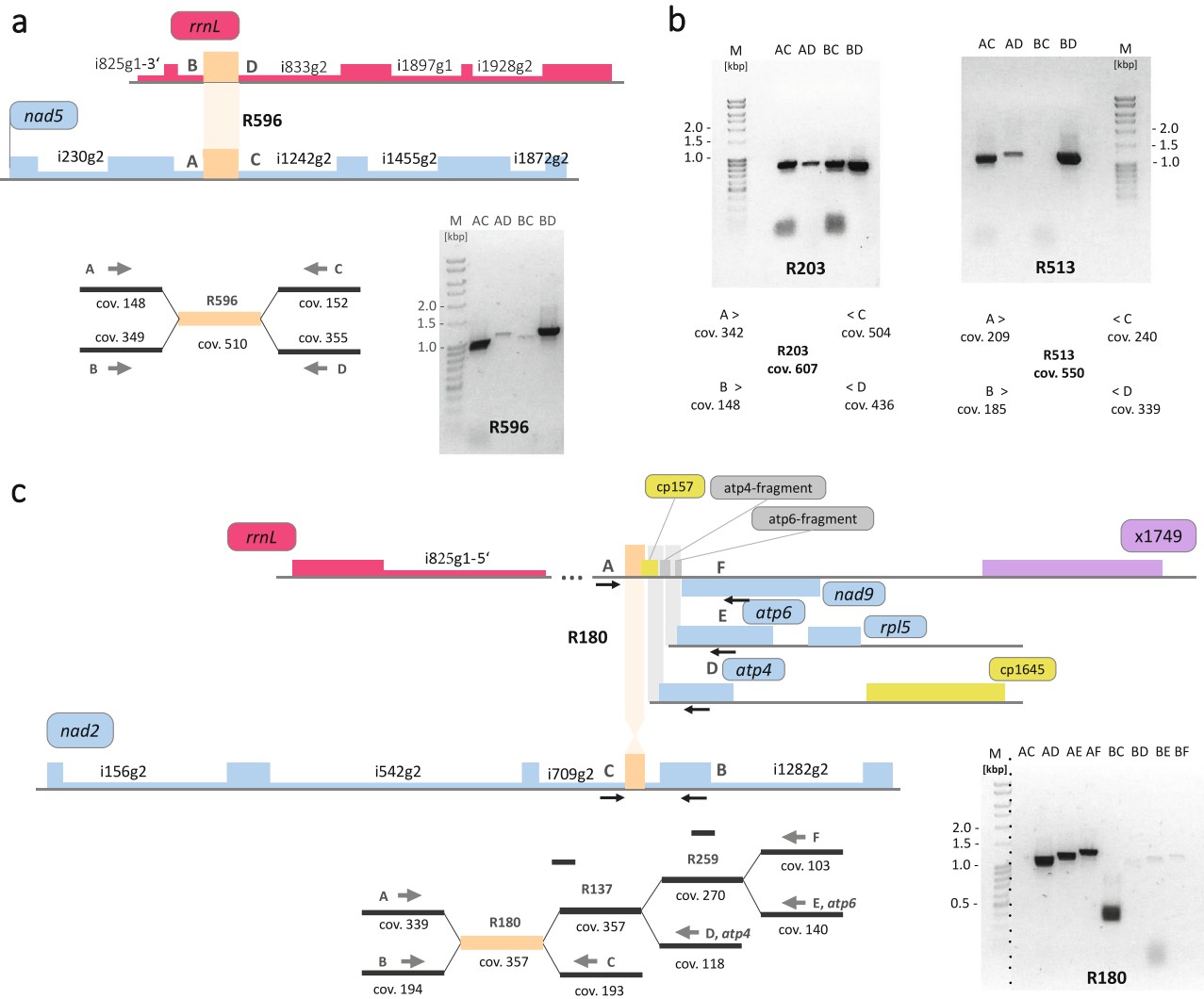

**Fig. 4 Repeats and recombination in the Haplopteris ensiformis mitogenome.** Recombinations across repeats (in orange) R596 (A), repeats R203 and R 513 (B) and R180 (C) was investigated by tsa-PCR ("template-switch-avoiding") strategies. **a** R596 is identically present in domains IV of group II introns nad5i1242g2 and rrnLi833g2. Average read coverage of the flanking single copy regions were ca. 150× for nad5 (arrangement A-C) and ca. 350× for rrnL (arrangement B-D), apparently adding up to ca. 500× for R596. PCR products are obtained for the expected gene continuities (AC, BD) with only minor evidence for reciprocal exchanges (AD, BC). **b** All combinations of flanking sequences (AC-BD) are identified for recombination across R203 whereas a clear bias is seen for recombination across R513 where a product for primer combination B-C remains undetected. **c** One copy of repeat R180 is located in intron nad2i709g2, another one downstream of the nad9 gene. The region between R180 and nad9 contains additional repeats R137 and R259 and all consecutive recombination products can be found whereas there is only very weak evidence for recombination across the R180 copy located in nad2i709g2. The marker lane of the electrophoresis gel covering R180 has been rearranged as indicated by the dotted line. The images of electrophoretic gels used in this figure are deposited as supplementary figures 5a and 5b.

nad5i1242g2 and rrnLi833g2, located on contigs H and F, respectively (Fig. 3). R596 had a coverage of ca. 500×, whereas the flanking single-copy contigs with nad5 and rrnL coding sequences had coverages of ca. 150× and 350×, respectively (Fig. 4a), indicating a non-equilibrium and different stoichiometries of the nad5 and rrnL gene copies. Naturally, active recombination would cause dysfunctional chimeric nad5/rrnL genes in this case. Exploring potentially active recombination across R596, we confirmed the gene continuities for nad5 and rrnL but found only very weak evidence for active recombination potentially giving rise to the two reciprocal arrangements (Fig. 4a).

However, similar tsa-PCR approaches run across other (intergenic) repeats indicated (stronger) active recombination across the respective repeated sequence resulting in more than only two co-existing, or at least strongly dominating,

conformations of the flanking single-copy sequences. Examples are shown for R203 and R513 (Fig. 4b). Whereas all combinations of flanking sequences appear to co-exist for R203, the arrangements A-C and B-D appear to strongly dominate for R513 while arrangement A-D is stoichiometrically under-represented and arrangement B-C even remains undetected (Fig. 4b).

As an additional complication, some recombination points at the end of repeats were in in very close proximity resulting in complex combined repeats. We here show results for R180, one copy of which is in group II intron nad2i709g2 of the nad2 gene (Fig. 4c). In another location, R180 is flanked on one side by consecutive repeats R137 and R259 resulting in alternative pathways to three different endpoints (D, E and F in Fig. 4c). Endpoints D and E result in intact atp4 and atp6 gene copies, respectively, whereas the alternative arrangement A-F creates pseudogene fragments for both genes downstream of nad9. Only

very weak evidence is found for recombination of nad2i709g2 across R180 into any of the three alternative arrangements B-D, B-E and B-F or for the reciprocal arrangement A-C (Fig. 4c).

Other than identical sequence repeats in distant locations, the mitogenome of *Haplopteris* carries copies of variably diverging sequences, sometimes closely spaced. Examples are an imperfect 39 bp IR embedding the *cox1* gene or an inverted sequence repeat of 1.1 Kbp embedding the *nad9* gene with (only) 70% sequence identity between the two copies. Direct repeats of an 85 bp sequence are located behind the *nad1* gene overlapped by a repeated tridecanucleotide motif (CCTCTACTGAGGG) at their ends.

**The mitochondrial gene complement in *H. ensiformis*.** Despite its highly complex structure, the *H. ensiformis* mitogenome has a rich gene complement (Table 1). All expected genes for subunits of the respiratory chain complexes I–V (*nad*, *sdh*, *cob*, *cox*, and *atp* genes) are present, including *sdh3* and *sdh4* encoding subunits of complex II. Likewise, there is a surprisingly large set of genes for ribosomal proteins, given that many are absent even in the mtDNAs of more ancestral lineages. Notable is the retention of *rps7*, which is lacking in all available hornwort and lycophyte mtDNAs but retained in *H. ensiformis*. Missing from the *Haplopteris* mtDNA is only the suite of *ccm* genes (*ccmB, ccmC, ccmF*) for cytochrome c maturation. Given their absence also in *Ophioglossum californicum* but their conservation in *Psilotum nudum* (Table 1) among the eusporangiate ferns, this evidently reflects a further independent loss of the *ccm* gene suite among ferns along with other phylogenetic deep losses in the lycophytes and hornworts.

The tRNA gene complement of the *H. ensiformis* mtDNA is particularly interesting owing to several tRNA genes of xenologous origin. Chloroplast-derived gene copies are present for trnF-GAA, trnMe-CAU, trnN-GUU, trnP-UGG and trnS-GGA (Table 1). Moreover, the *H. ensiformis* mtDNA also carries chlamydial-type tRNAs trnN-GUU (Fig. 3) and trnR-UCG (Figs. 2 and 3) described previously[20,29]. This results in xenologous genes coexisting with their native counterparts for trnF-GAA and trnP-UGG. Genes for native tRNAs are lost for trnL-CAA, trnMe-CAU, trnN-GUU, trnR-ACG. The chlamydial-type trnR-UCG is remarkable since it exists in three slightly differing copies. Similarly notable is the presence of a trnL-UAG in *Haplopteris* that is absent in the eusporangiate ferns (Table 1).

**RNA editing in *H. ensiformis* mitochondria.** Given the complex mitogenome structure, the parallel transcriptome and RNA editing analysis was fundamental to identify functional mitochondrial genes in the *H. ensiformis* mtDNA. All protein-coding genes were found to be affected by RNA editing. Altogether, we identified 1618 events of mitochondrial RNA editing: 1091 of the C-to-U type and 527 edits of the reverse U-to-C type (Supplementary Data 2). The abundance of RNA editing is highly biased among genes with the *cox1* mRNA being affected by 145 RNA editing sites in contrast to the *atp1* mRNA with only three edits, respectively.

While the very high coverage of RNA-Seq reads in chloroplasts allowed for determination of editing frequencies with high precision, the more than threefold abundance of edits in mitochondria allowed for a better statistical classification of edits categorized by their location and effect (Supplementary Data 2). Among the total of 1618 mitochondrial edits, 1171 introduce codon changes and of the latter more than 900 are strongly predicted and more than 100 others moderately or weakly expected. We here illustrate the prediction of editing sites by PREPACT[30] with the example of *atp9* finding a perfect match

between identified edits and expectations and for *atp6* with only minor deviations from the expected editing pattern (Supplementary Fig. 3). Notably, the strongly expected RNA editing events have an average editing frequency of 83%, much higher than only 53% on average for non-predicted changes of codon identities (Supplementary Data 2). Yet lower RNA editing efficiencies are observed for silent sites or the ones in non-coding regions, e.g. only 29% on average in 3'-UTRs (Supplementary Data 2). A notable exception from efficient editing of coding regions is the *matR* maturase encoded in the terminal *nad1* intron nad1i728g2 (Supplementary Data 2 and Table 1). Whereas identification of a start codon for this conserved, and only, mitochondrial group II intron maturase in flowering plants has been puzzling, we now find *matR* in *H. ensiformis* continuous with the upstream *nad1* reading frame, accordingly, to be labeled *mat-nad1i728g2c* following a recent nomenclature proposal[28]. Although only lowly edited, the numerous, and expected, events of RNA editing reconstituting conserved codons and including 15 stop codon removals confirm the functional role of *mat-nad1i728g2c* (Supplementary Data 2).

Particularly remarkable is that 44% of the reverse U-to-C edits (233 of 530) serve to re-convert stop codons into arginine or glutamine codons, an important issue to distinguish functional from dysfunctional pseudogenes. For example, within 20 codons upstream of rps3i249g2, six stop codons are removed from the *rps3* coding sequence, including one within the intron binding site (Supplementary Data 2).

A yet more dramatic example is the *rpl6-rps13-rps11* cluster with 20 genomic stop codons in a short region (Supplementary Fig. 4). In fact, the *rps11* gene at DNA level initially appeared to be an amino-terminally truncated pseudogene but turned out to have a proper start codon created by C-to-U editing and a total of seven stops removed by reverse U-to-C editing within the first 20 codons of its reading frame. We wished to test how different RT-PCR-based approaches would perform in comparison to the RNA-Seq approach to detect RNA editing sites. Towards that end, we used three different strategies for cDNA synthesis using either random hexamers or specific primers targeting the 3'-end of *rps11* in edited or non-edited versions. Here, we made use of two edits in the 3'-UTR closely behind the rps11eU466Q* stop codon editing. Sequencing of an internal amplicon revealed that many editing events were not confirmed in the cDNA sample primed with random hexamers with better performance by the specific primers and notably the one for the edited version of the 3'-end.

A striking bias concerns silent edits leaving codon identities unchanged. We observed only 32 silent U-to-C edits but the nine-fold amount (282) of silent C-to-U edits. Interestingly, silent C-to-U edits are frequently found to neighbor non-silent sites (NESIs) as has previously been observed for the huge editome in the *Selaginella uncinata* chloroplast[31].

**Mitochondrial introns in *H. ensiformis* include a *trans*-splicing group I intron.** The *H. ensiformis* mitogenome shows notable differences to the intron complements in the two eusporangiate fern taxa (Table 1). Ancient group II introns nad1i477g2, nad1i669g2, nad5i1477g2 and nad7i917g2 are lost from the *H. ensiformis* mtDNA. Vice versa, group I intron cox1i395g1 and group II introns cox2i373g2 and rps14i114g2 in the *Haplopteris* mitogenome lack counterparts in *Ophioglossum* and *Psilotum* (Table 1). The group II introns and cox1i395g1 are evidently of ancient origin in the land plant lineage. The latter has previously been identified in liverworts but also in the leptosporangiate ferns and in the horsetail *Equisetum arvense*[32,33].

A striking example documenting the *H. ensiformis* mitogenome complexity concerns the ribosomal rRNA cluster with a

**Table 1 The mitochondrial gene and intron complement of *Haplopteris ensiformis*.**

| Gene intron | *Psilotum nudum* | *Ophioglossum californicum* | *Haplopteris ensiformis* | Gene intron | *Psilotum nudum* | *Ophioglossum californicum* | *Haplopteris ensiformis* |
|---|---|---|---|---|---|---|---|
| atp1 | + | + | +[a] | rps1 | + | + | + |
| atp4 | + | + | + | rps1i25g2 | + | 0 | + |
| atp6 | + | + | + | rps2 | + | + | + |
| atp8 | + | + | +[a] | rps3 | + | + | + |
| atp9 | + | + | + | rps3i74g2 | + | + | + |
| ccmB | + | 0 | 0 | rps3i249g2 | + | 0 | + |
| ccmC | + | 0 | 0 | rps4 | + | + | + |
| ccmF | + | 0 | 0 | rps7 | + | + | + |
| ccmFCi829g2 | + | 0 | 0 | rps10 | + | + | +[a] |
| cob | + | + | +[a] | rps11 | + | + | + |
| cox1 | + | + | + | rps12 | + | + | + |
| cox1i395g1* | 0 | 0 | + | rps13 | + | + | + |
| cox1i624g2 | + | 0 | 0 | rps14 | + | + | + |
| cox2 | + | + | + | rps14i114g2* | 0 | 0 | + |
| cox2i373g2* | 0 | 0 | + | rps19 | + | + | + |
| cox3 | + | + | + | rrn5 | + | + | + |
| matR | + | + | + | rrnL | + | + | + |
| nad1 | + | + | + | rrnLi825g1* | 0 | 0 | T |
| nad1i258g2 | + | + | + | rrnLi833g2* | 0 | 0 | + |
| nad1i394g2 | + | 0 | + | rrnLi1897g1* | 0 | 0 | + |
| nad1i477g2* | + | + | 0 | rrnLi1928g1* | 0 | 0 | + |
| nad1i669g2* | + | + | 0 | rrnS | + | + | + |
| nad1i728g2 | + | + | + | sdh3 | + | + | + |
| nad2 | + | + | + | sdh4 | + | + | + |
| nad2i156g2 | + | + | + | tatC | + | + | +[a] |
| nad2i542g2 | + | + | + | trnA-ugc | + | + | +[b] |
| nad2i709g2 | + | + | + | trnC-gca | + | + | + |
| nad2i1282g2 | + | + | + | trnD-guc | + | + | + |
| nad3 | + | + | + | trnE-uuc | + | + | + |
| nad4 | + | + | + | trnF-gaa | + | + | + |
| nad4i461g2 | + | + | + | trnF-gaa cp* | 0 | 0 | +[c] |
| nad4i976g2 | + | + | + | trnG-gcc | + | + | + |
| nad4i1399g2 | + | + | + | trnG-ucc | + | + | + |
| nad4L | + | + | + | trnH-gug | + | + | +[b] |
| nad5 | + | + | + | trnI-cau | + | + | + |
| nad5i230g2 | + | + | + | trnK-uuu | + | + | + |
| nad5i1242g2 | + | 0 | + | trnL-caa | + | 0 | 0 |
| nad5i1455g2 | + | + | + | trnL-uaa[b,c] | + | + | + |
| nad5i1477g2* | + | + | 0 | trnL-uag* | 0 | 0 | + |
| nad5i1872g2 | + | + | + | trnMe-cau cp | + | + | +[c] |
| nad6 | + | + | + | trnMi-cau | + | Ψ | + |
| nad7 | + | + | + | trnN-guu cm* | 0 | 0 | + |
| nad7i140g2 | + | + | + | trnN-guu cp | + | + | +[c] |
| nad7i209g2 | + | 0 | + | trnP-ugg | + | + | + |
| nad7i676g2 | + | + | + | trnP-ugg cp* | 0 | 0 | +[c] |
| nad7i917g2* | + | + | 0 | trnQ-uug | + | + | + |
| nad9 | + | + | + | trnR-acg* | + | + | 0 |
| rpl2 | + | + | + | trnR-ucg cm | + | + | +[b] |
| rpl2i846g2 | + | + | + | trnR-ucu | + | + | + |
| rpl5 | + | + | + | trnS-gcu | + | + | + |
| rpl6 | + | + | + | trnS-gcu cp* | 0 | 0 | +[c] |
| rpl16 | + | + | + | trnS-gga cp* | 0 | 0 | + |
| | | | | trnS-uga | 0 | + | + |
| | | | | trnV-uac | 0 | + | + |
| | | | | trnW-cca | + | + | + |
| | | | | trnY-gua | + | + | +[b] |

List of mitochondrial genes and group I (g1) and group II (g2) introns in the *Haplopteris ensiformis* mitogenome in comparison to the ones of the eusporangiate ferns *Psilotum nudum* and *Ophioglossum californicum*. The added "cm" or "cp" indicate chloroplast-derived or chlamydial-type tRNA genes, respectively. Features distinguishing the *Haplopteris ensiformis* mtDNA from both eusporangiate ferns are tagged by an asterisk "*," i.e. the presence of a chlamydial *trnN-guu* gene, of chloroplast-derived *trnS-gcu* and *trnS-gga* genes, of introns cox1i395g1, rps14i114g2 and of four introns in the *rrnL* gene including the *trans*-splicing intron rrnLi825g1 vs. the absence of a *trnR-acg* gene and four other group II introns. The maturase *matR* in the terminal *nad1* intron, systematically labeled mat-nad1i728g2c, is in-frame with the upstream *nad1* coding region in *H. ensiformis*.
[a]Co-existing large pseudogene copies.
[b]Co-existing functional copies with only minor sequence differences.
[c]Chloroplast tRNA genes are part of extended cpDNA inserts.

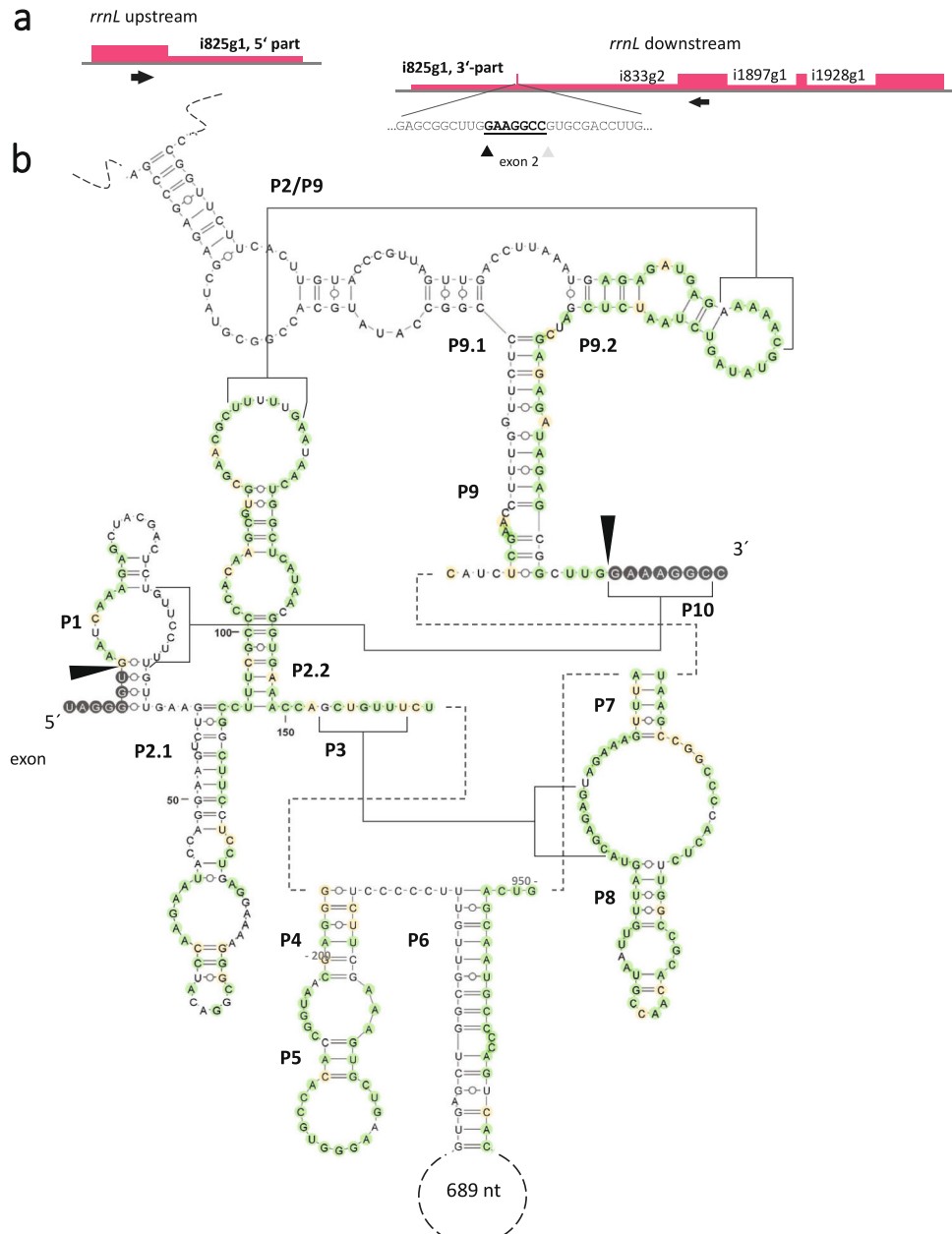

**Fig. 5 A trans-splicing group I intron in the Haplopteris ensiformis mitochondrial rrnL gene. a** Maturation of *rrnL* includes *trans*-splicing of the disrupted group I intron rrnLi825g1. Three additional introns are removed from the downstream part of *rrnL*. Group II intron rrnLi833g2 has a homolog conserved in the mtDNAs of liverworts and the downstream group I introns rrnLi1897g1 and rrnLi1928g1 are conserved in other Polypodiales species. Splice sites of rrnLi825g1 and rrnLii833g2 (black and gray triangles, respectively) frame the tiny second rrnL exon of only eight nucleotides. **b** Secondary structure model of the disrupted, *trans*-splicing group I intron rrnLi825g1. The figure was generated using the VARNA software[91]. Regions for base-pairing between the upstream and downstream parts of rrnLi825g1 are found in the group I intron secondary structure P9.1. Intron rrnLi825g1 intron has a positional ortholog in the sweet-water alga *Chara vulgaris*. Green and yellow shading indicates identical nucleotides and transitions, respectively.

peculiar arrangement featuring a disrupted *rrnL* gene (Figs. 2 and 3). Maturation of the large ribosomal RNA requires *trans*-splicing of a broken group I intron, rrnLi825g1. Secondary structure modeling suggests base-pairing interaction of the two intron parts in the disrupted domain P9.0/P9.1 (Fig. 5). Intriguingly, rrnLi825g1 has a distant positional ortholog as a *cis*-splicing homolog in the charophyte alga *Chara vulgaris*[34]. Despite overall only weak similarity, conserved regions include intron domains P7 and P8 known to contribute to the reactive core of group I introns. Three further introns are located in the downstream part of the *H. ensiformis rrnL* gene. Group II intron rrnLi833g2 is located only 8 bp downstream of the rrnLi825g1 3'-

splice site and has distant homologs in liverwort mtDNAs. The two downstream group I introns rrnLi1897g1 and rrnLi1928g1 have hitherto only been identified serendipitously in *rrnL* gene samplings of other Polypodiales species. Notably, none of the four introns are present in the eusporangiate ferns, which feature continuous *rrnL* genes.

**Laterally transferred chloroplast DNA fragments in the *Haplopteris* mitogenome.** Among other issues, the assembly of the *H. ensiformis* mitogenome was much complicated by the fact that it contains ca. 80 inserts of chloroplast DNA of variable sizes and with variable degrees of sequence conservation. Similar to the

repeat sequences, we annotated these cpDNA inserts with numbers indicating their sizes in base pairs preceded by "cp" (Supplementary Data 3). While the separate chloroplast sequence inserts may have originated from fragmentation after insertion of large stretches of cpDNA, their different degree of sequence conservation rather argue for independent transfer events and likely document independent cpDNA insertions at different time points in evolution (Fig. 6). As an example, an array of seven likely independently acquired cpDNA inserts is present in the intergenic region between *nad5* and *sdh4* (Fig. 6a). This region includes cp4165, the largest continuous stretch of promiscuous cpDNA derived from the *ndhH-ndhA-ndhI-ndhG-ndhE* region, sharing 93% sequence identity with the native chloroplast counterpart. This stretch is directly flanked by cp1271 derived from the chloroplast IR region encoding the *rrnL* gene and sharing even 99% of identical nucleotides, likely indicating a yet more recent inter-organellar migration into the mitogenome. Chloroplast insert cp1039 downstream of *rps4* (Fig. 6b) is another example for a likely recently acquired lateral sequence transfer. In a phylogenetic analysis, it branches next to the homologous sequence from the *atpA-atpF* region of the now determined *H. ensiformis* cpDNA as sister to the counterpart in *H. elongata*, even despite the generally high sequence drift among Pteridaceae (Fig. 6b).

At the other end of the spectrum, some cpDNA inserts lack detectable homologous sequences in the *Haplopteris* chloroplast genomes and could only be recognized by similarities with the cpDNA of other taxa. Inserts cp364 and cp749 (Fig. 6a) are examples, which could only be identified by their similarity to sequences in the IR regions in the cpDNAs of genera like *Asplenium* or *Vittaria*. Even more notable are the cases of morffos, the highly variable mobile ORFs in fern organelles[25]. The chloroplast sequence insert cp2126 is an example along those lines (Fig. 6c), which includes a morffo element that has a top sequence identity of 75% in the cpDNA of *Hemionitis subcordata* in the distant subfamily Cheilanthoideae.

Yet more striking is the case of morffo2, which is intact in the *H. elongata* cpDNA but truncated to its 5'-terminal 378 bp in the now determined *H. ensiformis* chloroplast genome (Fig. 1b). Intriguingly, chloroplast insert cp1712 in the mtDNA (Supplementary Data 3) includes remarkably more of the 5'-end of the truncated morffo2 element (628 bp). Taken together, the slow sequence drift in the mitogenome offers examples allowing for a molecular archeology of former chloroplast DNA sequences that are not present any more in the recent chloroplast genome. In the latter case, it further supports the point for degeneration of the morffo elements in *H. ensiformis* rather than their independent origin in *H. elongata* (Fig. 1b).

**Rickettsial DNA in the *H. ensiformis* mitogenome**. The chlamydial-type *trnN-GUU* gene initially identified in the lycophyte *P. squarrosus*[29] and the chlamydial-type *trnR-UCG* gene found subsequently in the eusporangiate fern mitogenomes[20] are now also identified in the *H. ensiformis* mtDNA, further corroborating the concept of HGT from bacteria into the mitogenomes of early-branching vascular plants. Most strikingly, we now discovered numerous inserts of "Rickettsia-like" sequences in the *H. ensiformis* mitogenome. Altogether, we identified 30 variably degenerated protein coding regions evidently derived from rickettsial bacteria (see Supplementary Data 3). Similar to the labels for repeats and cpDNA inserts, we annotated the xenologous bacterial DNA inserts indicating their respective extension in base pairs, in this case preceded by an "x" (Supplementary Data 3 and Fig. 7).

We carefully verified the surprising observation of Rickettsia-like DNA inserts in the *H. ensiformis* mitogenome by PCRs

anchoring in the flanking mtDNA regions and consistently corroborated the mitogenome assemblies, as we here exemplarily show for x625 representing a central coding region for the bacterial DNA recombination protein RmuC (Fig. 7a). Towards that end we used independent DNA preparations from two *H. ensiformis* isolates from separate locations and, vice versa, included material from *Vittaria lineata*, a closely related species that grows next to one of the *H. ensiformis* populations in the Botanic Garden Bonn. PCR products were consistently obtained for the two independent *H. ensiformis* isolates, but not for the *V. lineata* sample (Fig. 7a). Sequencing of the PCR products fully confirmed the mitogenome assembly, interspersed by Rickettsial-like DNA insertions. Moreover, none of the bacterial insert sequences pointed to contamination by living bacteria as they mostly revealed characteristic degeneration of the protein coding genes to pseudogenes. Finally, we observed continuities of the average read coverages continuing from flanking mitochondrial sequences into the bacterial DNA inserts (Fig. 7). We do not assume functional expression of the xenologous bacterial genes given that we could detect only negligible RNA coverage for some of the regions carrying bacterial DNA inserts.

After insertion into the *Haplopteris* mtDNA, the Rickettsia-type coding sequences degenerated by recombination and sequence drift. Some protein sequence similarities, however, remain astonishingly high, likely indicating very recent horizontal gene transfers (Supplementary Data 4 and 5). Specifically, top similarities were often observed with the corresponding sequences from *Caedimonas varicaedens*[35], followed by slightly lower similarities with homologous loci in *Cand. Paracaedimonas acanthamoebae*, other *Caedibacter* spp. or *Cand. Nucleicultrix amoebiphila*. These species belong to the family Holosporaceae among Rickettsiales *sensu lato*. The family Holosporaceae is alternatively considered to be a separate order of its own, the Holosporales.

As in the case of the chloroplast sequence inserts, the variable degrees of sequence degeneration suggest independent events of horizontal transfers at different time points in evolution. Alternatively, the insertion of larger xenologous genomic regions followed by later fragmentation in the mitogenome followed by different degrees of sequence degeneration may be possible. One evident example is the bacterial *HscA-RmuC* region (Fig. 7b) located between the *nad1* gene and cp232 (Fig. 7b, see Supplementary Data 3). The *HscA* coding region is disrupted by an insert in the coding sequence and the downstream *RmuC* gene is truncated with its central region located as x625 between the *cox2* gene and the *rrnL-rrnS* gene cluster, directly flanking cpDNA insert cp416 (Fig. 7a). Despite these discontinuities, the degree of amino acid sequence conservation reaching up to 94% identity is astonishing and phylogenetic analysis allows a close affiliation of the xenologous RmuC region in the *Haplopteris* mitogenome with *Caedimonas varicaedens* among the Holosporales (Fig. 7). Another xenologous insert x888 (Fig. 7b) carrying parts of the coding region for OMBB, an outer membrane beta-barrel domain containing protein, is much more degenerated and has much lower similarities to top-scoring hits with less clearly defined Rickettsiales bacteria (33% identity, 50% similarity).

Whereas most of the xenologous bacterial DNA inserts show fragmentation and considerable degeneration of protein-coding regions, there are also striking counterexamples like x1850, an insert of ca. 1.8 Kbp, containing the *serS-surE-nlpD* region located between cp763 and cp1456 (Supplementary Data 3). The coding region of the upstream serine tRNA ligase SerS is N-terminally truncated and carries two stop codons but the reading frame of the downstream coding region for the 5'/3'-nucleotidase SurE is perfectly conserved and shares even 97% sequence similarity with its *Caedimonas varicaedens* counterpart (Supplementary Fig. 2).

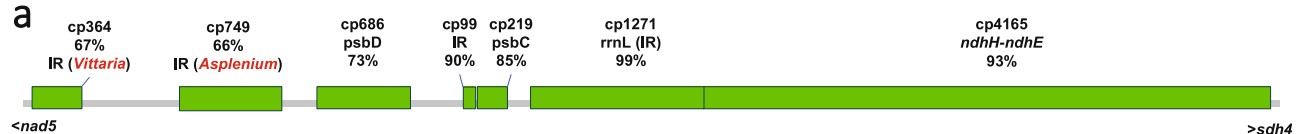

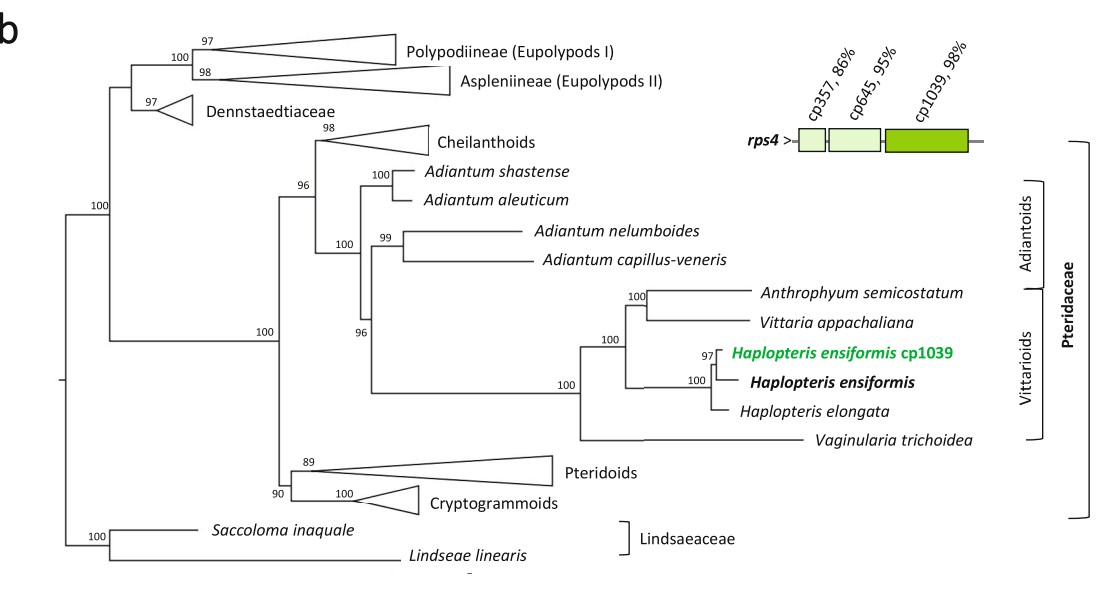

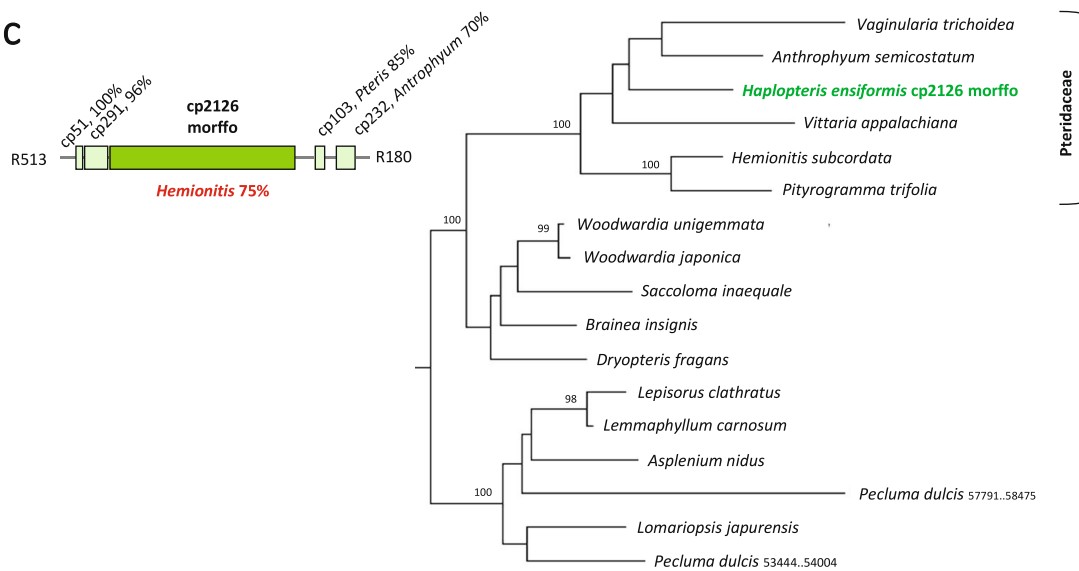

**Fig. 6 A multitude of chloroplast DNA inserts in the *Haplopteris ensiformis* mitogenome.** Selected examples for altogether approximately 80 inserts of chloroplast DNA populating the *Haplomitrium ensiformis* mitogenome (see Supplementary Data 3). Maximum likelihood trees were constructed with IQ-TREE [92] after automatic model selection of TIM+F+I+G4 or GTR+F+I+G4 and trees were rooted with the Lindsaeaceae family or the Eupolypod II clade, respectively, for cp1039 and cp2126. Bootstrap support is derived from 500 replicates. **a** The intergenic region between *nad5* and *sdh4* contains the largest collection of likely independently acquired cpDNA inserts including the largest individual insert cp4165 with 93% similarity to the native chloroplast *ndhH-ndhE* region. The other inserts share variable sequence identities with the native *H. ensiformis* cpDNA ranging from 73% for cp686 to 99% for cp1271. Inserts cp364 and cp749 lack evident homologies in the *H. ensiformis* plastome, but are identified by sequence similarities to cpDNAs in other fern genera like *Asplenium* or *Vittaria*, highlighted in red. **b** Chloroplast DNA insert cp1039 derived from the chloroplast *atpA-atpF* region is an example for a likely very recently acquired insert as evident from its well-supported sister placement to the newly assembled *H. ensiformis* cpDNA (Fig. 1) in a phylogeny including the homologous plastome regions from diverse polypod ferns. **c** Chloroplast insert cp2128 embedded in other cp inserts of variable sequence conservation carries a unique morffo element, identified by sequence similarity only in the cpDNA of *Hemionitis subcordata*.

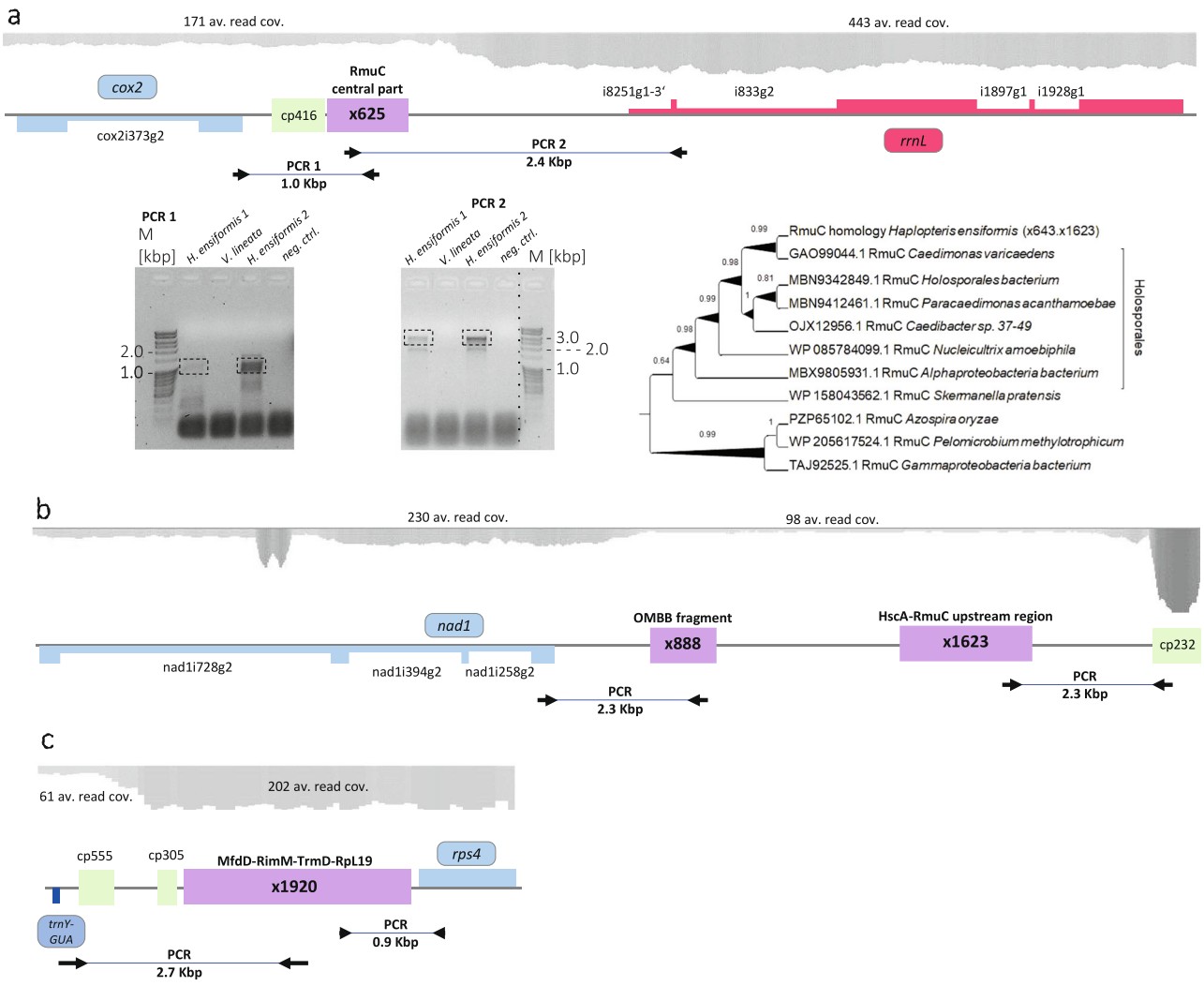

**Fig. 7 Rickettsial-like bacterial inserts in the *Haplopteris ensiformis* mitogenome.** Numerous inserts of bacterial, mostly rickettsial, protein-coding sequences were found to be integrated into the *Haplopteris ensiformis* mitogenome. Xenologous bacterial inserts were annotated to indicate their sizes in base pairs, preceded by "x" (see Supplementary Data 3), here showing examples for x625 (**a**), x888 and x1623 (**b**) and x1920 (**c**). Average read coverages are shown on top and PCR amplicons used to verify the mitogenome assemblies are indicated, with PCR results exemplarily shown for x625. **a** Bacterial insert x625 is located next to cpDNA insert cp416 between *cox2* and the downstream part of *rrnL*. Two overlapping PCRs confirm the mitogenome assembly with linkages into both genes for two independent samples from distantly grown *Haplopteris ensiformis* plant isolations but failed to find products for a *Vittaria lineata* sample growing near *H. ensiformis* isolate 1. PCR products of expected sizes (framed with stippled boxes) were cut out and sequenced and confirmed sequence identity with the mitogenome assembly. The graph on top shows a continuity for the average read coverages of ca. 170× for *cox2*, cp416 and x625 with an increase to ca. 440× for the downstream *rrnL* region. Insert x625 carries the central region for DNA recombination protein RmuC with the upstream part located on insert x1623. **b** Xenologous inserts x888 and x1623 are located between *nad1* and cpDNA insert cp232. Coding regions for HscA, an FeS-protein assembly chaperone and for RmuC borne on x1623 (see Supplementary Data 3 and Supplementary Data 4) are truncated, but highly conserved in primary sequence as exemplarily shown in the phylogenetic analysis for RmuC documenting a close association with *Caedimonas varicaedens*. Severely degenerated coding sequences for OMBB, an outer membrane beta-barrel domain containing protein borne on x888 are highly degenerated and do not allow a clear affiliation with a specific Rickettsiales bacterium. **c** Xenologous insert x1920 represents a continuous stretch of Rickettsia DNA with top similarities to four coding sequences in *Caedimonas* spp. Coding sequences (see Supplementary Data 4) are full-length for ribosome maturation factor RimM and the tRNA guanine-methyltransferase TrmD but amino-terminally truncated for MfdD, a transcription-repair-coupling factor and carboxy-terminally truncated for RpL19 encoding protein 19 of the large ribosomal subunit, respectively. The marker lane of the electrophoresis gel covering x625 PCR 2 has been rearranged as indicated by the dotted line. The original images of electrophoretic gels used in this figure are deposited as Supplementary Fig. 5c, d.

In this case a similarity of 81% can even be identified at the nucleotide sequence level (Supplementary Fig. 3), clearly suggesting the bacterial donor to be closely related to *C. varicaedens*. Intriguingly, the end of this bacterial sequence insert contributes to repeat R295.

Examples for particularly large stretches of Holosporaceae DNA inserts include x2170 derived from the bacterial XpsD-region located behind the mitochondrial *nad4L* gene (Supplementary Data 3)

and x1920, originating from the bacterial MfdD-RimM-TrmD-RpL19 region inserted upstream of *rps4* (Fig. 7c and Supplementary Data 3). In the latter case, the gene for MfdD is severely 5'-truncated, the gene for RimM appears intact, the TrmD reading frame is degenerated by a stop codon and the RpL19 sequence is 3' truncated (Fig. 7c).

In some cases, top sequence similarities clearly identify an origin of HGT sequences from Rickettsiales, but not necessarily

from within the Holosporaceae. Most notable is an assembly of sequence inserts of altogether more than 7 Kbp downstream of the *atp1* gene (Supplementary Data 3) that carries the suite of genes encoding murA and two CoA-carboxylase subunits in opposite direction, which are nearly unaffected by sequence degeneration except for in-frame stop codons (Supplementary Data 4). At nucleotide level, a top similarity of 74% identical nucleotides is observed with a not further identified Rickettsiaceae bacterium isolate PMG_002 (Supplementary Data 5). Intriguingly, the VirB8/B9 coding sequences of two P-type conjugative transfer proteins are located ca. 1.2 Kbp downstream of the murA homology. Top nucleotide sequence similarities are again observed for Rickettsiaceae bacterium isolate PMG_002. However, the virB8/B9 and the MurA coding sequences are not connected in that genome, indicating either a related and yet unidentified Rickettsiales donor, separate gene transfers or subsequent rearrangements after copy-transfer into the *Haplopteris* mtDNA.

Finally, two large stretches of protein coding regions are located upstream of *nad9*, running in opposite direction (labeled Bact-ORF 1 and 2, respectively), but cannot be assigned taxonomically owing to much lower similarities of only 40-50% at protein sequence level: A hypothetical ORF, possibly RNA polymerase, highest similarity with a *Magnetovibrio* sp. database entry (MBM08139) and a DNA-polymerase, highest similarity with a *Zoogloea* sp. sequence entry (KAB2964018).

**A complete leptosporangiate fern mitogenome assembly**. The large Pteridaceae family contains about 10% of extant ferns species[36–38]. Among them, the Adiantoid and Vittarioid subfamilies (see Fig. 6) show particularly high levels of substitution rate heterogeneity[39,40]. We became interested in the taxon given the apparently dynamic evolution of mitochondrial introns and RNA editing[22]. Likewise, a striking diversity of chloroplast RNA editing has been reported in the genus *Adiantum*[41].

The genomes of the two endosymbiotic organelles in *H. ensiformis* here reported are prime examples highlighting the discrepancy between the conservative evolution of chloroplast DNA and the highly dynamic evolution of vascular plant mitochondrial genomes. Among the recently reported flowering plant mitogenomes, the one of the holoparasite *Ombrophytum subterraneum*[42] is a case in point documenting not only a multi-chromosomal structure but multiple evidence for HGT from its host plants. Like in other cases of multi-chromosomal plant mitogenomes, e.g., the hundreds of different mitochondrial chromosomes in some *Silene* species, no efforts are made any more to come up with the display of a hypothetical, and likely misleading, master circle[43,44].

As a representative for the large clade of leptosporangiate ferns with ca. 10,000 species, the *H. ensiformis* mitogenome again adds to the list of astonishing molecular peculiarities that are hallmarks of vascular plant mtDNAs[45,46]. Given its extraordinary complexity, it comes as no surprise that no other leptosporangiate mitogenome has previously been assembled despite multiple NGS efforts including the water fern genera *Azolla* and *Salvinia*[47,48] or the flying spider-monkey tree fern *Alsophila spinulosa*[49] or, most recently, the model ferns *Adiantum capillus-veneris*[50] and *Ceratopteris richardii*[51]. Likewise no mitogenome was reported in a genome assembly effort for the lycophyte *Isoetes taiwanensis*[52], possibly owing to even higher complexity than the ones reported previously for *Isoetes engelmannii* and *Selaginella moellendorffii*[7,8].

We hence speculate that highly complex mitogenomes like the one reported here may be a general feature of leptosporangiate ferns also outside of the Polypodiales. Intriguingly though, and

despite its highly dynamic structural evolution, the *H. ensiformis* mtDNA contains a surprisingly rich set of classic mitochondrial genes (Table 1) when compared to the gene complement of other taxa including the eusporangiate ferns[53].

**Mitochondrial intron dynamics in ferns**. More notable than the gene complements are the diverging mitochondrial intron complements now identified in *H. ensiformis* in comparison to the previously analyzed eusporangiate ferns, extending earlier conclusions that much more intron dynamics is present in monilophytes in comparison to their seed plant sister clade[21,22,33,53]. The *Haplopteris* mitogenome reveals retention of evidently ancient introns that have been lost in the eusporangiate ferns like cox1i395g1, cox2i373g2, rps14i114g2 and rrnLi833g2 (Table 1).

Surprisingly though, the heavily recombining mtDNA of *H. ensiformis* has not resulted in disrupted group II introns like in angiosperms or in gymnosperms where ever more transitions to *trans*-splicing have been observed recently[54]. In contrast, most group II introns (11 of 15) found to be *trans*-splicing in at least one seed plant lineage (cox2i373g2, nad1i394g2, nad1i728g2, nad2i542g2, nad2i1282g2, nad4i461g2, nad4i976g2, nad4i1399g2, nad5i1455g2, nad7i209g2 and rpl2i846g2) are present in conventional *cis*-arrangements in the *Haplopteris* mtDNA fully in line with the early evolutionary conclusion that *trans*-splicing introns in seed plants originate from *cis*-arranged ancestors in early-branching plant lineages[55,56]. We speculate that transitions from *cis*- to *trans*-splicing group II introns may rely on co-evolving protein splicing factors that are present in the seed plant but not in the monilophyte lineage. The remaining four group II introns known to exist in a *trans*-splicing state in at least some spermatophytes (cox2i691g2, nad1i669g2, nad5i1477g2 and nad7i917g2) have been lost altogether from the *H. ensiformis* mitogenome (Table 1).

In the light of the above, it is all the more surprising to find rrnLi825g1 as a group I intron of yet unclear ancestry in a *trans*-splicing arrangement in the idiosyncratic *rrnL* gene makeup in the *H. ensiformis* mitogenome. In contrast to the numerous examples of *trans*-splicing group II introns in plant organelles alone, *trans*-splicing group I introns appear to exist much more rarely in nature. First reports of *trans*-splicing group I introns in the *Isoetes engelmannii* mitogenome[7] and in the mtDNA of *Trichoplax*[57] have been followed by recognition of *trans*-splicing group I intron cox1i744g1 in *Helicosporidium* mtDNA[58] and of two *trans*-spliced group I introns in *Gigaspora margarita* mtDNA[59]. Remarkably, the *trans*-splicing group I intron cox1i395g1 in the *Isoetes engelmannii* mtDNA[7] exists in a conventional, *cis*-arranged version in the *H. ensiformis* mitogenome.

The mitochondrial *rrnL* gene is entirely devoid of introns not only in seed plants and lycopyhtes but also in hornworts[28] and even in the eusporangiate ferns[20], whereas it features four introns in the *H. ensiformis* mtDNA: rrnLi825g1, rrnLi833g2, rrnLi1897g1 and rrnLi1928g1 (Fig. 5). The latter two group I introns have already been documented serendipitously in sequence samplings covering parts of the mitochondrial *rrnL* gene[60] and are conserved also outside of Pteridaceae in taxa as least as distant as tree ferns (Cyatheales, e.g. *Plagiogyria stenoptera*, accession DQ647877). The origins of these introns remain unclear as they neither share sequence similarities anywhere else in the plant lineage nor among fungi as is the case for the rampant invader group I intron cox1i726g1 sporadically occurring in angiosperms[61,62].

**Verifying native and foreign sequences in the organelle genomes**. The examples of the *H. ensiformis* organelle genomes

reported here document that parallel analysis of transcriptome along with genome NGS sequencing data is essential to characterize genes as functional or dysfunctional owing to the complex maturation processes including C-to-U and U-to-C RNA editing and *cis*-splicing or *trans*-splicing of split genes and introns. Moreover, the highly complex mitogenome of *H. ensiformis* is a prime case showing that very careful investigations of alien sequences in NGS data to tell them apart from native chloroplast DNA or bacterial contaminations may be needed in such complex cases of organelle genome structures. Aside from our experimental verifications, we note that parts of rickettsial DNA inserts in *H. ensiformis* also seem to be present in database entries reporting partial mtDNA sequences of the ferns *Asplenium nidus* (FR669448) and *Dryopteris crassirhizoma* (MW732172). Moreover, we suggest the careful re-evaluation of some fern cpDNA entries (like *Dipteris conjugata* KP136829, *Polypodium vulgare* MT984517, *Cystopteris protrusa* KP136830 or *Selliguea yakushimensis* MN623352) that seem to contain mtDNA stretches as possible artifacts. Certainly, however, once verified and when the likely similarly complex mitogenomes of those taxa would be assembled in the future they may document interesting inter-organellar DNA transfer from mitochondria to chloroplasts.

### Lateral sequence transfers: cpDNA insertions in the *Haplopteris* mitogenome.
The first evidence for lateral transfer of promiscuous chloroplast DNA into a plant mitogenome has already been documented 40 years ago[63] and thereafter found to be merely a standard feature in many seed plant mtDNAs. Most interestingly, among the numerous cpDNA insertions in the mtDNA of *H. ensiformis* we now find examples evidently documenting ancient cpDNA features that even are not present any more in the recent plastomes, as here seen for the mysterious morffo elements. The origin and dynamics of these only recently described mobile ORFs in fern organelles[25] is presently still enigmatic. Here, we find that apparently intact morffos in the cpDNA of *H. elongata* have evidently disintegrated in the plastome of *H. ensiformis* (Fig. 1b) but that their counterparts and evidence for yet other morffos is present in its mitogenome as evolution's mis-placed witnesses.

### Origins of bacterial sequences in the *Haplopteris* mtDNA.
The role of horizontal gene transfers (HGT) is increasingly appreciated not only for bacterial evolution but also in the evolution of eukaryotic genomes including ferns[64,65]. In several cases, host-parasite interactions are key to the events of HGT[42,66] and such interactions in nature may also be responsible for gene transfer into fern mtDNA[67]. In particular, after first reports on horizontal plant-to-plant transfer of mtDNAs[68,69], it is meantime well understood that HGT has contributed to many seed plant mitogenomes[70]. The most outstanding example is the case of the early-branching flowering plant *A. trichopoda* having integrated into its mitogenome not only numerous stretches of mtDNA from other angiosperms but also the near-complete mitogenomes of two mosses[19,71]. A new dimension of HGT into plant mitogenomes opened up with the discovery of tRNA genes from chlamydial origins into very early tracheophyte lineages[29]. Yet more recently, it was found that sequences of fungal origin have been horizontally transferred early into mitogenomes of the Orchid family[72].

Here, we now report on a multitude of Rickettsia-like genome insertions as one of the most astonishing findings emerging from the assembly and analysis of the *H. ensiformis* mitogenome. Intriguingly, Rickettsiales are known to be obligate intracellular parasites. Their phylogenetic relatedness to the progenitor of the eukaryotic mitochondrion is a matter of ongoing debate[73,74]. The bacterial DNA inserts in the *H. ensiformis* mtDNA are most likely derived from species closely related to (Cand.) *Caedibacter acanthamoebae*[35] of the Holosporales (or, alternatively, Holosporaceae among Rickettsiales). *Caedibacter* (or *Caedimonas*) endosymbionts transfer the killer trait to their *Paramecium* hosts[75,76]. Other Rickettsia are known to be associated with arthropods, leeches and protists and Rickettsia-like organisms (RLOs) and are associated not only with human or animal diseases but also with numerous plant diseases, for example the Rickettsia endosymbiont of the tobacco whitefly *Bemisia tabaci*. Moreover, Rickettsia have also been associated with a papaya disease[77] and have been identified eustigmatophyte algae[78] and in the green alga *Mesostigma viride*[79]. It will be very interesting to see whether Rickettsial or related bacterial DNA insertions will also be identified in further leptosporangiate mitogenomes and to ultimately identify the exact donor species and the biological mechanisms of the HGT processes. Notably, the xenologous bacterial DNA regions identified in the *H. ensiformis* mitogenome include similarities to the IS481 family transposase and virB8 and virB9 homologs (Supplementary Data 3). The latter genes are commonly found among mobile IS elements and are associated with conjugative gene transfer amongst Rickettsia species[80].

### Conclusions
The leptosporangiate fern mitogenome reported here shows that routine pipelines for organelle genome assemblies are prone to fail when mitogenomes become as complex as in the case of the *H. ensiformis* mitochondrial DNA. Notably, mitogenomes were not produced during genome assemblies from NGS data obtained recently for several ferns and a lycophyte, as mentioned above. It will be highly interesting to see whether our findings will help to carefully re-consider and re-analyze the NGS data in those cases and whether lateral and horizontal gene transfers will have affected mitogenomes and complicated their assemblies to similar or possibly even higher degrees. The presence of foreign sequences arising from lateral transfer of cpDNA or horizontal transfer of bacterial DNA will demand careful re-investigation and additional experimental confirmation similar to the approaches that we have reported here. Further discoveries of rickettsial, chlamydial or possibly sequences of yet other bacterial origins will motivate to explore the physical interactions in nature of the donor bacteria and fern prothallia as the likely entry points for the horizontally transferred sequences. Similarly intriguing is the surprising retention of *cis*-arranged group II introns in the highly recombinant mitogenome of *H. ensiformis*, possibly indicating that only intron-binding proteins present in seed plants but not in fern nuclear genomes have allowed for the *cis*-to-*trans* transition of their ribozymic structures.

### Methods

**Plant material.** The Bonn University Botanic Garden kindly provided plant material for *H. ensiformis* (xx-0-BONN-24687) and *V. lineata* (xx-0-BONN-17295). Species identities were independently verified by PCR amplification and sequencing of the *rbcL* and *atpA* locus revealing complete sequence identities with independent sequence accessions (*H. ensiformis* KX164999 and MH359250 and *V. lineata* EF473712 and KU744782).

**Organelle genome sequencing and assembly.** The Qiagen DNeasy Plant Mini Kit was used for DNA isolation and the Sigma-Aldrich plant RNA isolation Kit for RNA preparation followed by ribosomal RNA depletion using the RiboMinus Plant Kit for RNA-Seq (Thermo Fisher Scientific). RNA quality was checked with an Agilent 2100 Bioanalyzer system for RIN values of at least 8.5. Genome and transcriptome sequencing (paired-end whole genome sequencing and RNAseq) was done commercially at the BGI on an Illumina platform. After further quality control of DNA and RNA samples at the BGI, paired-end reads of 150 and 100 nt length were produced on the Illumina HiSeq X Ten and Hiseq 2500 platform, respectively. Raw read data were evaluated with FastQC v0.11.9

(http://www.bioinformatics.babraham.ac.uk/projects/fastqc/). No adapter artifacts could be detected for 31,091,834 clean RNA reads with a mean sequence quality of 36 (>99.9% base call accuracy) for forward and reverse reads (Supplementary Data 6). The MEGAHIT v1.2.9 software[81,82] was used for *de novo* whole genome assembly with independent runs using three different settings ("strict", "default" and "relaxed"). The "strict" assembly was run with the minimum starting k-mer size and small number of increments of k-mer size iterations (--k-min 21 --k-step 16). The third assembly was run with parameters set for a high ultra-complex metagenomics dataset (--k-min 27 --k-step 10). Complete assembly of cpDNA was performed with NOVOPlasty 2.3.2[83] using contigs from the MEGAHIT assembly with conserved chloroplast genes as seeds. The sequencing and assembly statistics for our study are available in Supplementary Data 6. This file contains information on the quality control measures taken after sequencing, as well as details on the assembly of the organellar genomes and the identification of RNA editing sites.

RNA reads were assembled with the Trinity v2.8.2 software[84,85]. The BLAST 2.9.0+[86] suite was used to initially identify contigs with chloroplast or mitochondrial gene content using available lycophyte and fern organelle genomes. Raw read data and assembly data have been deposited under BioProject accession no. PRJNA862965. Whole genome assembly contigs could be clustered into two sets based on k-mer coverage (mt/cp 1:10). After connecting, contigs were verified by PCRs (see below), contigs of the mitochondrial genome were connected by hand.

**Verification of mtDNA arrangements.** PCRs were used to independently verify the highly complex arrangements of the *H. ensiformis* mitochondrial DNA resulting from multiple repetitive sequences and insertions of chloroplast and xenologous DNA. PCR amplicons were designed with primers anchoring in neighboring sequence regions of evident mitochondrial identity, preferably coding regions. Special care was taken to investigate repeated sequences for potential recombination creating alternative arrangements of flanking sequences. To best avoid false positives suggesting active recombination resulting from artificial template switches we used a strategy of template-switch avoiding tsa-PCRs. To that end, a mix of gel-eluted PCR fragments containing a repeat sequence in different sequence environments (AB and CD) was used to obtain products reflecting a reciprocal exchange of flanking sequences (AD and CB). A series of template dilutions (1:10, 1:20, 1:30, 1:40, 1:50 and 1:60) and numbers of PCR cycles (15, 20, 25 and 30) were tested and adjusted to determine the threshold for artificial production of template-switch products.

Identification of mitochondrial genes avoided routine pipelines but started from homologs in the mtDNA of diverse taxa, including the liverwort *Marchantia polymorpha*, the lycophyte *P. squarrosus* and the gymnosperm *Ginkgo biloba*. Gene identities were verified by the parallel transcriptome studies to confirm intron splicing and C-to-U and U-to-C RNA editing. Identification of tRNA genes combined the use of tRNAscan-SE[87] and sensitive BLASTN searches using a tRNA query set including the recently identified chlamydial tRNA xenologues in early tracheophytes[20,29]. To identify DNA similarities including repeats, chloroplast DNA or xenologous bacterial DNA insertions, we used sensitive BLASTN or XBLAST similarity searches (word sizes = 7 or 3, respectively) and strict random expectancy threshold cutoffs of 1e−10. On nucleotide level, this translates approximately into identification of identical repeated sequences of ca. 40 bp (i.e. slightly larger than the conserved domain V of group II introns) or respective larger, but less similar, regions.

The primers used in our PCR experiments on the verification of mtDNA arrangements, transcriptome studies, and the determination of Rickettsial DNA insertions within the mtDNA of *H. ensiformis* can be found in Supplementary Data 7.

**Transcriptome studies and determination of RNA editing sites.** Transcriptome studies were used to determine all intron splicing sites. The identification of RNA editing sites was done as previously described[28]. Briefly, DNA and RNA reads were mapped against the organelle contig sequences using GSNAP[88] and JACUSA[89] was used to determine RNA-DNA differences. Thresholds were set to minimally 30 reads and RNA editing efficiencies of at least 1% for chloroplast and at least 5% for mitochondrial transcripts for a strict determination of C-to-U and U-to-C RNA editing sites, respectively. Selected loci were analyzed independently by RT-PCR-based cDNA analyses and sequencing for various reasons like unexpectedly inefficient RNA editing at certain sites or because of coexisting pseudogene copies as discussed under results (chloroplast *rpoC1* and mitochondrial genes *atp1, atp8, nad5, nad7, rrnL* and the *rpl6-rps13-rps11* co-transcript).

**Statistics and reproducibility.** *H. ensiformis* DNA and RNA for Illumina sequencing were prepared from the same isolate from the Botanic Garden of the University Bonn, accession 24687. PCR experiments were conducted on the same samples sent in for Illumina sequencing and in parallel for at least one other specimen of *H. ensiformis* accession 24687 taken from a different location in the Botanic Garden.

**Reporting summary.** Further information on research design is available in the Nature Portfolio Reporting Summary linked to this article.

## Data availability

*Haplopteris ensiformis* primary nucleotide sequence reads are submitted to the sequence read archive (SRA) under BioProject accession number PRJNA862965. The assembled chloroplast genome is deposited under accession number OM867544 and the assembled mitogenome chromosomes are available under accession numbers OM867545–OM867553. Oligonucleotide sequences for PCR experiments are provided in Supplementary Data 7. Uncropped versions of the electrophoretic gels shown in Figs. 4c and 7a have been deposited as Supplementary Fig. 5.

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

## Acknowledgements

The authors wish to thank the Molecular Evolution group for stimulating discussion and productive interaction, especially Philipp Gerke for his contributions to establish pipelines for the analysis of RNA editing in NGS data. The authors gratefully acknowledge the support for this project by computing time provided by the Paderborn Center for Parallel Computing (PC²). Finally, we thank the staff at the Botanical Garden of the university Bonn for providing materials and their kind support.

## Author contributions

S.Z. did wet lab work and established bioinformatic pipelines. M.P. helped with nucleic acid preparations and molecular cloning. S.Z. and V.K. analyzed data and prepared figures. V.K. wrote the manuscript and all authors edited and approved the final manuscript version.

## Funding

## Competing interests

The authors declare no competing interests.
