## [Peer review File · Communications Biology]

Reviewers' comments:

Reviewer #1 (Remarks to the Author):

This manuscript reports the sequencing and analysis of the mitochondrial and plastid genomes of the leptosporangiate fern *Haplopteris ensiformis*. The study includes an exceptional detailed description of genomic features, including HGT, RNA editing, gene and intron content, and recombinational activity. All of these are well-established features of plant mitogenomes, so the study does not radically add to our understanding of plant organelle genomes. But the analysis appears to be unusually thorough and rigorous, and the contribution fills a relatively large taxonomic gap in the availability of plant mitogenomes. In such clades where plant mitogenomes are highly unstable, each one truly is a snowflake with its own unique elements. Perhaps, the most interesting observation in this study was the extensive and apparently recent transfer of DNA from a bacterial donor into the mitogenome. As far as I can tell, the analysis and conclusions here appear to be sound, so I have only very minor comments.

1. In the summary, the authors should consider clarifying what they mean that the mitogenome is "lacking only the ccm gene suite." Lacking only those genes relative to what? Perhaps the ancestral set in land plants? Certainly this species lacking genes that have been retained in the mitogenomes of some eukaryotes outside of land plants.

2. Check for consistent usage of mitogenome and chondroma. I did not see a need to interchange both of these terms.

3. Lines 559-560. The authors state that Rickettsiales are "assumed to be the extant alpha-proteobacterial lineage most closely related to the progenitor of the eukaryotic mitochondrion." I do not think this currently represents a consensus in the field. In fact, there has been evidence recently that mitochondria may even represent a lineage that is sister to all extant alphaproteobacteria rather than one that is actually nested within alphaproteobacteria. I would suggest the authors more clearly acknowledge the current uncertainty in the field.

4. Lines 585-586. I am confused by the description of the RNA QC. To my knowledge, RIN scores are produced by a Bioanalyzer or a TapeStation ("RINe" in that case). The authors state that they produced a RIN score with a Qubit. The Qubit 4.0 has recently introduced a kit to estimate RNA integrity. I am not very familiar with this application of a Qubit, but it appears to generate something called an RNA IQ score rather than a RIN score. Another point of confusion is that the authors state that scores were >0.9. I believe all the aforementioned scores range from 1 to 10 (with scores expected to be on the high end of that range for suitable sequencing quality). I suggest clarifying the basis of RNA QC.

5. For both the sequencing and bioinformatic methods, I suggest providing a bit more detail, including sequencing read lengths, the specific type of Illumina instrument that was used, and details on software versions and parameters used in analysis. I feel that such details would aid in reproducibility.

Reviewer #2 (Remarks to the Author):

Haplopteris ensiformis, commonly known as the shoestring ferns, is a species of epiphytic fern. In green plants (Viridiplantae), research effort has been preferentially concentrated in plastids; thus, the majority of the diversity of mitogenomes awaits further exploration. Plant mitogenomes can be difficult to assemble because they are structurally dynamic and prone to intergenomic DNA transfers, leading to the unusual situation where an organelle genome is far outnumbered by its nuclear counterparts. In the present study, the authors present the organelle genomes of *Haplopteris ensiformis* as the first leptosporangiate fern. I found the manuscript to be interesting and well-written. The objective of the paper is clear and unambiguous. English language and style of the whole text are correct with rather minor drawbacks. However, before accepting the manuscript in its current form, I have several concerns and would suggest the following major/minor revisions:

- 1.The authors used the second-generation data to assemble the organelle genome. But I couldn't find the sequencing and assembly stats. This is very much important to judge the quality of the data and the interpretations made using such data.
- 2.There are nine main figures, I think this can be reduced to 5-6, depending on the journal requirements.
- 3.Discussion lack a clear conclusion and future perspectives of the study.

Dear reviewers,

thank you very much for letting us have the kind and positive reviews on our submission and the helpful comments in detail to improve our paper. Below, we provide a detailed list of responses (bold) to the points raised by the reviewers (italics) along with a correction-tracked version of our manuscript.

Reviewer #1 (Remarks to the Author):

*This manuscript reports the sequencing and analysis of the mitochondrial and plastid genomes of the leptosporangiate fern *Haplopteris ensiformis*. The studied includes an exceptional detailed description of genomic features, including HGT, RNA editing, gene and intron content, and recombinational activity. All of these are well-established features of plant mitogenomes, so the study does not radically add to our understanding of plant organelle genomes. But the analysis appears to be unusually thorough and rigorous, and the contribution fills a relatively large taxonomic gap in the availability of plant mitogenomes. In such clades where plant mitogenomes are highly unstable, each one truly is a snowflake with its own unique elements. Perhaps, the most interesting observation in this study was the extensive and apparently recent transfer of DNA from a bacterial donor into the mitogenome. As far as I can tell, the analysis and conclusions here appear to be sound, so I have only very minor comments.*

1. In the summary, the authors should consider clarifying what they mean that the mitogenome is "lacking only the ccm gene suite." Lacking only those genes relative to what? Perhaps the ancestral set in land plants? Certainly this species lacking genes that have been retained in the mitogenomes of some eukaryotes outside of land plants.

We agree that more clarity is needed here. We have reworded the sentence to "The *Haplopteris* mtDNA is gene-rich, lacking only the *ccm* gene suite ancestrally present in early land plant mitogenomes..."

2. Check for consistent usage of mitogenome and chondroma. I did not see a need to interchange both of these terms.

OK. We have replace the two occurrences of "chondrome" with "mitogenome" in lines 38 and 65.

3. Lines 559-560. The authors state that Rickettsiales are "assumed to be the extant alpha-proteobacterial lineage most closely related to the progenitor of the eukaryotic mitochondrion." I do not this currently represents a consensus in the field. In fact, there has evidence recently that mitochondria may even represent a lineage that is sister to all extant alphaproteobacteria rather than one that is actually nested within alphaproteobacteria. I would suggest the authors more clearly acknowledge the current uncertainty in the field.

Given the reviewer's comment, we have now cited two papers on the subject (new references 72 and 73, respectively) reflecting a very recent discussion of the subject and accordingly reworded the text (lines 559-562) as follows: "Intriguingly, Rickettsiales are known to be obligate intracellular parasites. Their phylogenetic relatedness to the progenitor of the eukaryotic mitochondrion is a matter of ongoing debate^{72,73}."

4. Lines 585-586. I am confused by the description of the RNA QC. To my knowledge, RIN scores are produced by a Bioanalyzer or a TapeStation ("RINe" in that case). The authors state that they produced a RIN score with a Qubit. The Qubit 4.0 has recently introduced a kit to estimate RNA integrity. I am not very familiar with this application of a Qubit, but it appears to generate something

called an RNA IQ score rather than a RIN score. Another point of confusion is that the authors state that scores were >0.9. I believe all the aforementioned scores range from 1 to 10 (with scores expected to be on the high end of that range for suitable sequencing quality). I suggest clarifying the basis of RNA QC.

We apologize for the confusion caused by re-arranging text sections before submission (Qubit fluorometry was, in fact, another quality check later performed at the BGI. We have reworded the sentence to now read: (lines 602-3): “RNA quality was checked with an Agilent 2100 Bioanalyzer system for RIN-values of at least 8.5.”

5. For both the sequencing and bioinformatic methods, I suggest providing a bit more detail, including sequencing read lengths, the specific type of Illumina instrument that was used, and details on software versions and parameters used in analysis. I feel that such details would aid in reproducibility.

We apologize for not having provided that information, which was, however, included with the details in the BioProject accession PRJNA862965 (see lines 619-20). In line with the comments by reviewer #2, we have now added this also under methods, now reading (lines 605-14):

“After further quality control of DNA and RNA samples at the BGI, paired-end reads of 150 and 100 nt length were produced on the Illumina HiSeq X Ten and HiSeq 2500 platform, respectively. Raw read data were evaluated with FastQCv0.11.9 (<http://www.bioinformatics.babraham.ac.uk/projects/fastqc/>). No adapter artifacts could be detected for 31,091,834 clean RNA reads with a mean sequence quality of 36 (>99.9% base call accuracy) for forward and reverse reads. The MEGAHIT v1.2.9 software^{80,81} was used for *de novo* whole genome assembly with independent runs using three different settings (“strict”, “default” and “relaxed”). The “strict” assembly was run with the minimum starting k-mer size and small number of increments of k-mer size iterations (--k-min 21 --k-step 16). The third assembly was run with parameters recommended for a high ultra-complex metagenomics dataset (--k-min 27 --k-step 10).”

In addition, we added version numbers to the software packages named in the material and methods section: NOVOplasty2.3.2 (line 615) BLAST2.9.0+ Trinity v2.8.2 (line 617)

Reviewer #2 (Remarks to the Author):

Haplopteris ensiformis, commonly known as the shoestring fern”, is a species of epiphytic fern. In green plants (Viridiplantae), research effort has been preferentially concentrated in plastids; thus, the majority of the diversity of mitogenomes awaits further exploration. Plant mitogenomes can be difficult to assemble because they are structurally dynamic and prone to intergenomic DNA transfers, leading to the unusual situation where an organelle genome is far outnumbered by its nuclear counterparts. In the present study, the authors present the organelle genomes of Haplopteris ensiformis as the first leptosporangiate fern. I found the manuscript to be interesting and well-written. The objective of the paper is clear and unambiguous. English language and style of the whole text are correct with rather minor drawbacks. However, before accepting the manuscript in its current form, I have several concerns and would suggest the following major/minor revisions:

1. The authors used the second-generation data to assemble the organelle genome. But I couldn't find the sequencing and assembly stats. This is very much important to judge the quality of the data and the interpretations made using such data.

This criticism is in-line with comment 5 by reviewer #1. The details are now provided with the new methods text section in the revised manuscript as outlined above.

2. There are nine main figures, I think this can be reduced to 5-6, depending on the journal requirements.

In response to this comment we have decided to move figures 2 and 6 addressing cp and mt RNA editing into the supplementaries (as new Supplementary Figures S1 and S4, respectively). However, the remaining figures illustrate important issues of the complex organelle genome makeups and lateral sequence transfers and are integral to the key messages of our paper. Hence, we would prefer to keep those seven figures among the primary illustrations. The re-numbering of figures and illustrations is correction-tracked in the revised manuscript version.

3. Discussion lack a clear conclusion and future perspectives of the study.

While we had tried to address the respective issues separately among the sub-headlines of the discussion, we agree that an overall “Conclusions and perspectives” paragraph will enhance the key messages. Accordingly, we have added a new final paragraph to the discussion reading: “Conclusions and perspectives

The first leptosporangiate fern mitogenome reported here shows that routine pipelines for organelle genome assemblies are prone to fail when mitogenomes become as complex as in the case of the *Haplopteris ensiformis* mitochondrial DNA. Notably, mitogenomes were not produced, during genome assemblies from NGS data obtained recently for several ferns and a lycophytes, as mentioned above. It will be highly interesting to see whether our findings will help to carefully re-consider and re-analyze the NGS data in those cases and whether lateral and horizontal gene transfers will have affected mitogenomes and complicated their assemblies to similar or possibly even higher degrees. The presence of foreign sequences arising from lateral transfer of cpDNA or horizontal transfer of bacterial DNA will demand careful re-investigation and additional experimental confirmation similar to the approaches that we have reported here. Further discoveries of rickettsial, chlamydial or possibly sequences of yet other bacterial origins will motivate to explore the physical interactions in nature of the donor bacteria and fern prothallia as the likely entry points for the horizontally transferred sequences. Similarly intriguing is the surprising retention of cis-arranged group II introns in the highly recombinant mitogenome of *H. ensiformis*, possibly indicating that only intron-binding proteins present in seed plants but not in fern nuclear genomes have allowed for the *cis-to-trans* transition of their ribozymic structures.”

Again, we wish to express our gratitude for the friendly reviews and the helpful comments by the two reviewers to further improve our paper. We hope that you will find the changes and amendments now introduced to adequately address the reviewers’ remarks and that you will find the revised manuscript version suitable for publication.

With kind regards,

Volker Knoop and Simon Zumkeller, also on behalf of Monika Polsakiewicz

Reviewers' comments:

Reviewer #2 (Remarks to the Author):

First of all, I appreciate that authors have nicely revised the manuscript. However, my one comment is still not addressed. I still couldn't find the sequencing and assembly stats in the revised version.

Dear reviewers,

thank you very much for the feedback on our revised manuscript. To address the remaining point on sequencing and assembly stats we have added further text and two further supplements as detailed below. Again, we include a correction-tracked version of our manuscript.

Reviewer #2 (Remarks to the Author):

First of all, I appreciate that authors have nicely revised the manuscript. However, my one comment is still not addressed. I still couldn't find the sequencing and assembly stats in the revised version."

Comment in first review

"1. The authors used the second-generation data to assemble the organelle genome. But I couldn't find the sequencing and assembly stats. This is very much important to judge the quality of the data and the interpretations made using such data."

Response:

We have now added Supplementary Material 1, which includes assembly statistics from fastqc quality control, organellar genome assemblies, and mapping for SNP-calling experiments to identify RNA editing sites. While we are not investigating a nuclear genome in this study and the organellar genomes only include single-copy genes, we do not use BUSCO for control of "completeness". Our results indicate that the CP gene complement is complete and the MT gene complement is surprisingly large with the exception of the *ccm* gene suite, which was previously shown to be lost independently (see results section, lines 250 ff.). For the same reason, we cannot provide numbers on unmapped reads.

In addition, we have included a complete list of primers used in the numerous PCR experiments involved in the assembly and verification of the mtDNA in Supplementary Material 2.

We address the additions in the manuscript text as follows in lines 608, 614-617 and 648-650:

"The sequencing and assembly statistics are available in Supplementary File 1. This file contains information on the quality control measures taken after sequencing, as well as details on the assembly of the organellar genomes and the identification of RNA editing sites."

"The primers used in our PCR experiments on the verification of mtDNA arrangements, transcriptome studies, and the determination of Rickettsial DNA insertions within the mtDNA of *H. ensiformis* can be found in Supplementary File 2."

We hope that you will find our manuscript suitable for publication with the above outlined additional amendments and wish to thank you again for all the efforts related to our submission.

With kind regards and best wishes for the new year,

Simon Zumkeller and Volker Knoop

REVIEWERS' COMMENTS:

Reviewer #2 (Remarks to the Author):

The manuscript can be accepted now.